# Structural insights into the functional mechanism of the ubiquitin ligase E6AP

Zhen Wang [1,8], Fengying Fan[1,2,8], Zhihai Li[2,3,8], Fei Ye [4,8], Qingxia Wang[5], Rongchao Gao[5], Jiaxuan Qiu[1,2], Yixin Lv[2,3], Min Lin[4], Wenwen Xu[1,2], Cheng Luo [1,2,3,6,7] ✉ & Xuekui Yu [1,2,5] ✉

E6AP dysfunction is associated with Angelman syndrome and Autism spectrum disorder. Additionally, the host E6AP is hijacked by the high-risk HPV E6 to aberrantly ubiquitinate the tumor suppressor p53, which is linked with development of multiple types of cancer, including most cervical cancers. Here we show that E6AP and the E6AP/E6 complex exist, respectively, as a monomer and a dimer of the E6AP/E6 protomer. The short α1-helix of E6AP transforms into a longer helical structure when in complex with E6. The extended α1-helices of the dimer intersect symmetrically and contribute to the dimerization. The two protomers sway around the crossed region of the two α1-helices to promote the attachment and detachment of substrates to the catalytic C-lobe of E6AP, thus facilitating ubiquitin transfer. These findings, complemented by mutagenesis analysis, suggest that the α1-helix, through conformational transformations, controls the transition between the inactive monomer and the active dimer of E6AP.

Protein ubiquitination is carried out by a three-enzyme cascade consisting of the ubiquitin-activating enzyme (E1), the ubiquitin-conjugating enzyme (E2), and the ubiquitin ligase (E3)[1–3]. E3 ligases catalyze the last step of this cascade by transferring ubiquitin from the E2 enzyme to the target protein(s). E3 ligases are categorized into three major classes: RING (really interesting new gene), RBR (ring-between-ring), and HECT (homologous to the E6AP carboxyl terminus) ligases[4–6]. The HECT-type E3 ligases have a characteristic C-terminal HECT domain consisting of an N-lobe and a C-lobe that are responsible for E2 binding and ubiquitin acceptance, respectively. During ubiquitination, the ubiquitin moiety is first transferred from a specific E2 enzyme to the active-site cysteine of the C-lobe, and then to the lysine(s) of the substrate[7–10]. Each HECT-type E3 ligase has a distinct amino terminus that drives substrate recognition, E3 activity regulation, and ubiquitin-linkage specificity[7–9].

Human protein E6AP/UBE3A, the founding member of the HECT-type ubiquitin ligase family, was originally identified as an associated partner of the viral oncoprotein E6 of human papillomaviruses (HPVs)[11,12]. The E6 oncoprotein from high-risk HPV types hijacks the ligase activity of the host E6AP to ubiquitinate the tumor suppressor p53[13], leading to the aberrant p53 degradation. This aberrant degradation of p53 allows the virus-infected cells to evade apoptosis and continue proliferating, thereby contributing to the development of multiple HPV-positive cancers, including over 99% of cervical cancers[14–21]. Rescuing the functionality of p53 is thus an appealing strategy against HPV-positive tumors[22,23]. Immunotherapy targeting E6 has been conducted in clinical trials for the treatment of HPV-associated cancers[24,25].

Previous studies have shown that neither E6AP nor E6 alone can bind with p53[11], and that an 'LxxLL' motif (where x is any amino acid) located at the amino terminus of E6AP is essential for E6 binding[26–28].

[1]State Key Laboratory of Drug Research, Shanghai Institute of Materia Medica, Chinese Academy of Sciences, Shanghai 201203, China. [2]University of Chinese Academy of Sciences, Beijing 100049, China. [3]School of Pharmaceutical Science and Technology, Hangzhou Institute for Advanced Study, University of Chinese Academy of Sciences, Hangzhou 310024, China. [4]College of Life Sciences and Medicine, Zhejiang Sci-Tech University, Hangzhou 310018, China. [5]Cryo-Electron Microscopy Research Center, Shanghai Institute of Materia Medica, Chinese Academy of Sciences, Shanghai 201203, China. [6]Zhongshan Institute for Drug Discovery, Shanghai Institute of Materia Medica, Chinese Academy of Sciences, Zhongshan 528437, China. [7]Department of Pharmacy, Guiyang University of Traditional Chinese Medicine, South Dong Qing Road, Huaxi District, Guizhou 550025, China. [8]These authors contributed equally: Zhen Wang, Fengying Fan, Zhihai Li, Fei Ye. ✉e-mail: cluo@simm.ac.cn; xkyu@simm.ac.cn

Indeed, structural studies have provided the mode of interaction between an LQELL-containing peptide (E6AP peptide) and E6[27,29]. However, due to a lack of structural knowledge about most of the N-terminal regions and the HECT catalytic domain of E6AP in these structures, it is still unclear how the E6AP/E6 full-length complex assembles and how the ligase activity of E6AP is exploited by E6 to ubiquitinate p53.

In addition to cancer, E6AP has been linked with neurodevelopmental disorders including Angelman syndrome and Autism spectrum disorder[30–37]. Angelman syndrome is caused by either the loss of maternal *E6AP* or E6AP loss-of-function missense mutations[30,33,34]. In contrast, Autism spectrum disorder has been linked with duplication of maternal *E6AP* or gain-of-function missense mutations of E6AP[31,33–37]. These studies suggest the requirement for tight physiological control over E6AP function.

Here, by using cryogenic electronic microscopy, we reveal that the structures of full-length E6AP and E6AP/E6 complex are monomer and dimer of E6AP/E6 protomer, respectively. We show that the α1-helix of E6AP is extended in the E6AP/E6 complex and makes extensive contributions to the formation of the dynamic dimer of the E6AP/E6 protomer. We identify a pocket for E6 binding in the E6AP of E6AP/E6 tetramer. The dynamicity of the E6AP/E6 complex, as exhibited by the resolved five conformational states, favors proximity of the E6AP's C-lobe to E2 for accepting ubiquitin or to substrates for donating ubiquitin. Our site-directed mutagenesis, coupled with structural analyses, suggest that dimerization of the E6AP/E6 protomer is essential for p53 ubiquitination. These results reveal that conformational transformation of the α1-helix changes E6AP from an inactive monomer to an active dimer, and thereby provides the long sought-after structural understanding of the physiological and pathophysiological mechanisms of E6AP function.

## Results

### A dimer of the E6AP/E6 protomer

High-quality samples of the E6AP/E6 complex were obtained through co-expression of the full-length E6AP and E6 of HPV16 (Supplementary Fig. 1a). A total of 21,152 cryoEM movies were collected with a 300 kV Titan Krios microscope and over 16 million particles were picked for analysis (Supplementary Fig. 1b–d). Both the 2D averages and the 3D initial model of the E6AP/E6 complex indicated structural features with 2-fold symmetry (Supplementary Fig. 1c, d). Thus, we applied C2 averaging in the subsequent 3D classifications and autorefinements. After several rounds of 3D classification, we identified five distinct conformational states of the E6AP/E6 complex and resolved the structures of these five conformations at resolutions ranging from 2.6 to 4.4 Å (Supplementary Fig. 1e and Supplementary Table 1).

We used the 2.6-Å density map to build the ab initio atomic model (Fig. 1a–e and Supplementary Fig. 2). The overall structure of the E6AP/E6 complex looks like two leaning "towers" standing on a base (Fig. 1a, c, d). Each tower derives exclusively from E6AP's N-terminal region (residues 120-169, 231-384, and 427-483), whereas the base consists of the C-terminal HECT domain (residues 484–875), the loop-helix fragment, and the LxxLL-containing loop-helix-loop element (residues 385-426) stretching down from the tower region (Fig. 1d). The structure of E6 in the E6AP/E6 protomer comprises the E6 N-terminus (E6N), the E6 C-terminal (E6C) zinc-binding domains, and a linker helix tethering the E6N to the E6C (Fig. 1e); this E6 linker helix is almost identical to the counterparts of the E6AP peptide/E6 complex as well as the E6AP peptide/E6/p53 core domain complex resolved previously[27,29].

We noted that a 14-residue loop-helix fragment located at the inter-protomer interface was connected neither to E6AP nor E6 (Fig. 1d, f, g). Given that the E6 model was missing only 8 and 9 residues at its N- and C-termini (Fig. 1e), respectively, this fragment did not belong to E6. E6AP, on the other hand, has two missing regions: the

first comprises an N-terminal 119 residues, while the second spans residues 170 to 230 (Fig. 1d). We assigned the fragment to the missing N-terminal region of E6AP based on two facts: 1) the two ending residues of 169 and 231 are far away from the loop-helix fragment; and 2) these residues were located on opposite sides of the tower and, therefore, the 47 missing residues did not appear to be long enough to allow both residue 169 and residue 231 to connect with the loop-helix fragment. Following the principle of proximity, we tentatively assigned the identities of the two loop-helix fragments and named each of them as the N-helix. It is worth noting that these assignments are for the convenience of the structural description only, and our discussions and conclusions remain independent of these assignments. Using the atomic model derived from the 2.6-Å density map, we then built the models for the other four conformations of the E6AP/E6 complex (Supplementary Fig. 3a–c). All five conformations of the E6AP/E6 complex present as a dimer of the E6AP/E6 protomer.

While the overall structure of each E6AP/E6 protomer is similar among the five conformations (Supplementary Fig. 3d), the spatial relationship between the two protomers varies. We therefore assigned the five conformations into one of two classes: attached (Att) or detached (Det). In all three Att conformations, the catalytic C-lobe of E6AP attaches to E6 (Fig. 1a–c and Supplementary Fig. 3a–c), whereas, in the two Det conformations, there are no interactions between the C-lobe and E6 (Supplementary Fig. 3a-c). We hereafter refer to the three Att conformations as Att1, Att2 and Att3 based on the distance between the tips of the two towers in a descending sequence: 74 Å in Att1, 71 Å in Att2, and 66 Å in Att3 (Fig. 1c and Supplementary Fig. 3b). Following the same nomenclature, the Det two conformations are referred to as Det1 (76 Å) and Det2 (61 Å) (Supplementary Fig. 3b). Unless otherwise specified, we use the Att1 conformation to describe the structural features and molecular interactions herewith.

### Inter-protomer interactions of the E6AP/E6 complex

Only the zinc-binding E6N and E6C domains interact with the C-lobe of E6AP (Fig. 2a, b). Residues H31 and D32 of E6N form hydrogen bonds with T834 and R836 of the C-lobe, respectively. Residue H31 also contacts R836 of the C-lobe via a cation-pi interaction, and residues R46, R47, and Y50 of E6N interact with N830, P832, H841, and L848 of the C-lobe through van der Waals forces (Fig. 2c). R148 of E6C simultaneously interacts with E775 and D777 of the C-lobe through electrostatic interactions, and S145 of E6N forms a hydrogen bond with D777 of the C-lobe (Fig. 2d).

Since the interactions between the C-lobe and E6 do not exist in the Det conformations, these interactions are likely to be dispensable in terms of the dimerization of the E6AP/E6 protomer. Instead, dimerization relies on the interaction between the two 3-helix bundles, which are found in all five conformations (Fig. 1f, g and Supplementary Fig. 3c). Each of these 3-helix bundles is formed by an α1-helix (termed as the first helix in the HECT domain, residues 486-514), the N-helix of E6AP, and the E6 linker helix. The two E6 linker helices are not involved in the dimerization interactions. While the two N-helices have weak end-to-end contacts (Fig. 1f), the two α1-helices have strong hydrophobic interactions (Fig. 2e), which contribute significantly to the dimerization of the two E6AP/E6 protomers. Consistently, the E6AP N-terminal truncation mutant (residues 120-870) can still be co-purified with E6 (Supplementary Fig. 4a); the static light-scattering (SLS) analysis showed that the molecular weight of the purified E6AP mutant/E6 complex corresponding to the major elution peak (~7.5 mL) was characteristic of a dimer of the E6AP mutant/E6 protomer (Supplementary Fig. 4b). Also of note, the E6AP mutant has a similar capacity as the wild-type E6AP to ubiquitinate p53 (Supplementary Fig. 4c, d).

### Dynamics of the E6AP/E6 complex and ubiquitin transfer

The structural changes among the five states could be classified into two types of motion between the two protomers of the E6AP/E6

 

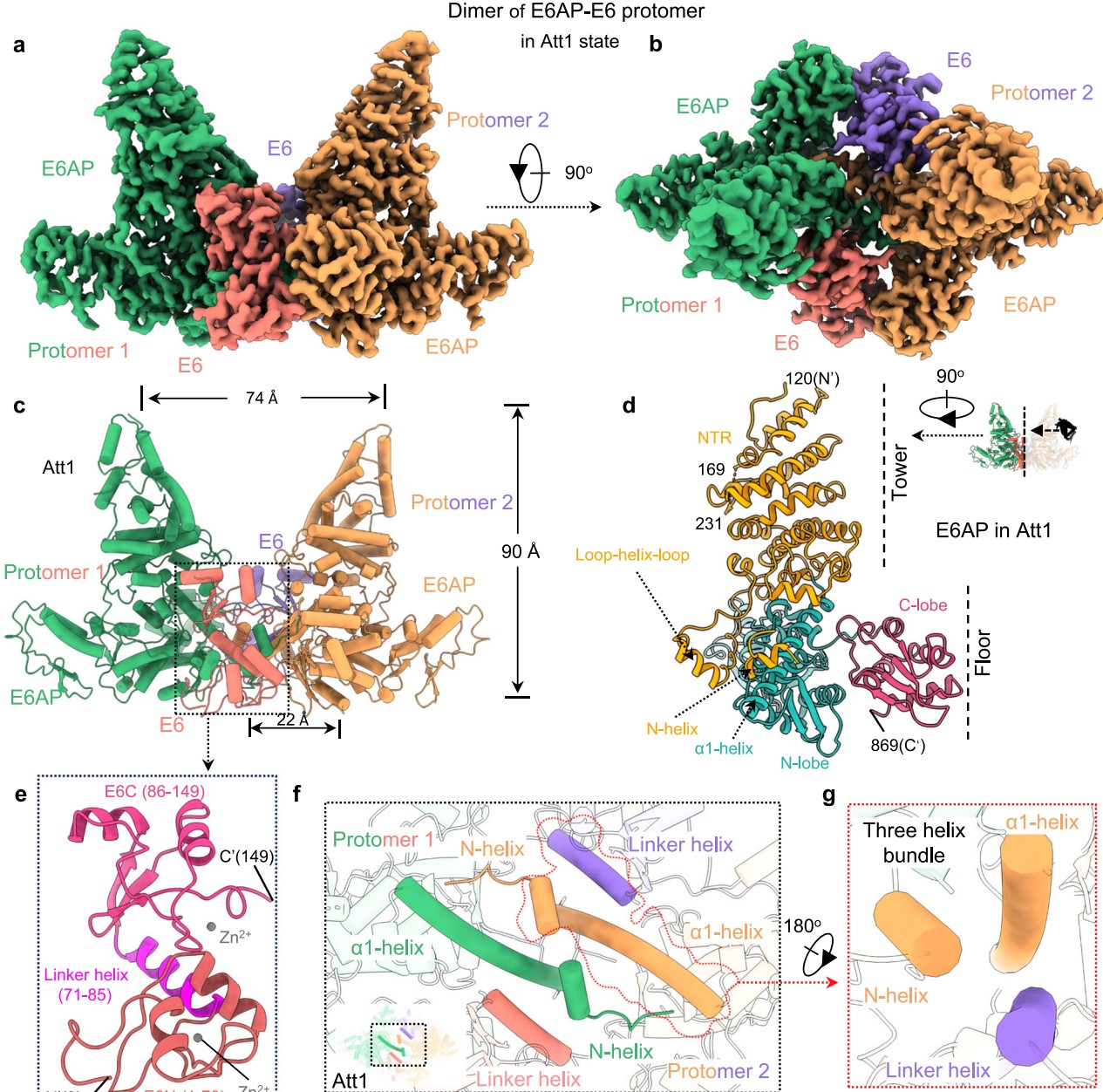

**Fig. 1 | Overall structure of the E6AP/E6 complex. a, b** Side view (**a**) and top view (**b**) of the density map of the E6AP/E6 complex. The E6AP/E6 complex is a dimer of the E6AP/E6 protomer in the attached-1 (Att1) state. E6AP and E6 are green and red, respectively, in protomer 1 and yellow and purple, respectively, in protomer 2. Maps were processed using DeepEMhancer. **c** Atomic model of the E6AP/E6 complex. The view and colors are same as that in **a. d** Structural architecture of E6AP in the E6AP/E6 complex. E6AP presents as a tower standing on a base. The tower is exclusively produced from the N-terminal region (NTR, yellow) of E6AP, whereas the base is formed by contributions from the C-terminal HECT domain (green N-lobe and red C-lobe), the N-helix, and the loop-helix-loop element. Note that the structural features presented here are based on the conformation of the E6AP/E6 complex in the Att1 state. **e** Structural characteristics of E6 in the E6AP/E6 complex in the Att1 state. E6 contains an E6N domain, an E6C zinc-binding domain, and a linker helix tethering E6N to E6C. **f** Two 3-helix bundles at the interface between two E6AP/E6 protomers in Att1 state. The small cartoon in the lower left corner is shown for orientation purposes and indicates a close-up view enclosed by the black line. E6AP and E6 are colored as in **b. g** Zoomed-in view of the regions outlined by the red line.

complex (Fig. 3a–d). (1) the two towers tilt towards each other around a region of intersect between the two α1-helices (Fig. 3a, b). As such, the conformation of E6AP/E6 complex changes from Att1, through Att2 to Att3. To better understand this, we applied 3D variability analysis (3DVA) using particles in the three attached states (approximately 419,000 particles, Supplementary Fig. 1d). The result was a series of conformations of the complex that depicted continuous movement between the two protomers (Supplementary video 1). The conformational changes among the three Att states are essentially similar to that

revealed by 3DVA but with compromised continuum. (2) When the two towers leant beyond a threshold, the Att conformation changed into a Det conformation. As the two towers underwent an outwards rotation, the Det2 conformation transformed into the Det1 conformation (Fig. 3c, d). Intriguingly, despite substantial variation in the structure of the E6AP/E6 complex, the inter–α1-helical interaction remains intact (Fig. 1f and Supplementary Fig. 3c); albeit there are some changes at the interface between the two α1-helices (238 Å² for Att1, 222 Å² for Att2, 209 Å² for Att3, 157 Å² for Det1 and 145 Å² for Det2).

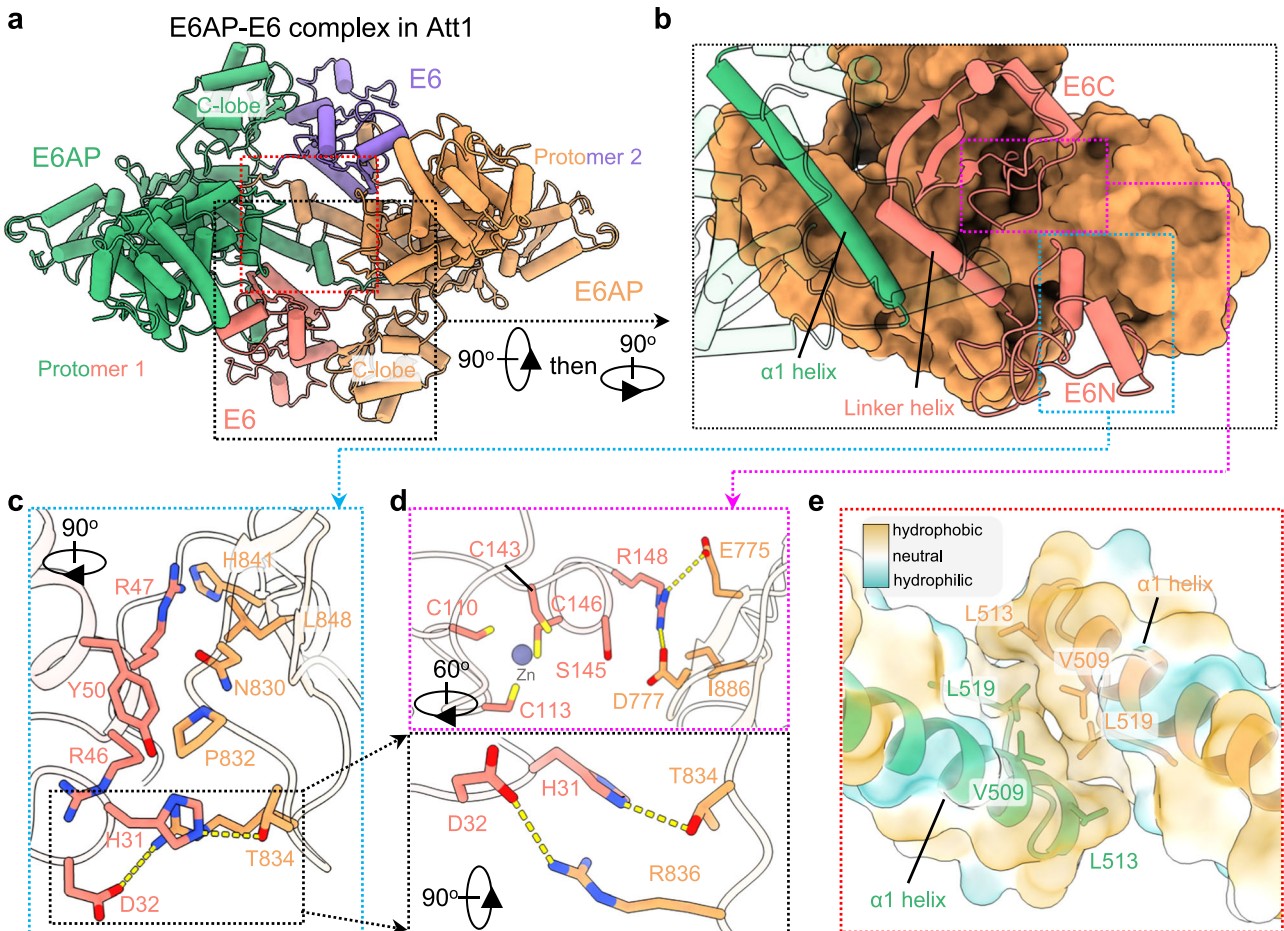

**Fig. 2 | Inter-protomer interactions of the E6AP/E6 complex. a** Overall interaction between two E6AP/E6 protomers in the Att1 state. The inter-protomer interactions are mainly mediated by the α1-helix and the N-helix (red box) and by the E6 and C-lobe of E6AP (black box). The orientation, the coloring and the method of naming are the same as that presented in Fig. 1b. **b** Zoomed-in view of the inter-protomer interaction between E6 and the C-lobe. E6AP in the protomer 2 is shown in surface representation. **c, d** Detailed interactions between the two E6AP/E6 protomers mediated by E6 and the E6AP C-lobe. The interactions are formed mainly by E6N (blue box, **c**) and E6C (red box, **d**) of E6 in protomer 1 with the C-lobe of E6AP in protomer 2. **e** Zoomed-in view of the regions outlined by the red line in **a**. Lipophilicity potential surface (hydrophobic, gold; hydrophilic, cyan) and ribbon presentations show the detailed interactions between two E6AP/E6 protomers mediated by the α1-helix.

The E6 viral oncoprotein is a substrate of E6AP[38]. Accompanying the E6AP/E6 complex conformational changes, there is substantial variation in the spatial relationship between the E6 from one protomer and the catalytic C-lobe of E6AP of the other protomer: far away (Det1), close (Det2), or in contact (Att), as described above (Fig. 1c and Supplementary Fig. 3b). In contrast, E6 is maintained far away from the intra-protomer C-lobe, regardless of the conformational state of the E6AP/E6 complex (Supplementary Fig. 5); indeed, the E6 and the intra-protomer E6AP C-lobe are located on opposite sides of the NTR (Fig. 1b) and are therefore inaccessible to each other. These structural insights indicate that dimerization of the E6AP/E6 protomer is essential for E6 ubiquitination.

While the distance between the two towers varies among the three Att conformations, there is no detectable difference in the distance between the C-lobe and the inter-protomer E6 (Fig. 1c and Supplementary Fig. 3b). It is likely that Att conformational changes of the E6AP/E6 complex are needed to appropriately position the E6AP C-lobe for ubiquitin donation to E6. It is also plausible that the interactions between the C-lobe and E6 contribute to this process. To examine this, we constructed an E6 mutant (D32A/Y50A/R148A) bearing mutations at distinct sites of the E6/C-lobe interface (Fig. 2c, d) to attenuate the interaction between the C-lobe and E6, and conducted a ubiquitin transfer assay. We found that the E6 mutant-bound E6AP

transferred less ubiquitin to substrates including itself and E6 as compared to the wild-type E6-bound E6AP (Fig. 3e, f). On one hand, the E6 mutant-bound E6AP has more ubiquitin linked to its catalytic site (Fig. 3e). On the other hand, the E6 mutant-bound E6AP and the E6 mutant show less ubiquitination level (Fig. 3f and Supplementary Fig. 6). These results suggest that interactions between the E6AP C-lobe and E6 contribute to for the donation of ubiquitin from E6AP to its substrate.

We superimposed the five conformations of the E6AP/E6 complex with the crystal structure[29] of the E6AP peptide/E6/p53 core domain (residues 94–292) via alignment of the Cα of E6 to analyze the role of the dimerization of the E6AP/E6 protomer in p53 ubiquitination. Among the five conformations, only the Det1 conformation showed the docked p53 core domain as being away from the inter-protomer C-lobe (Supplementary Fig. 7a); the other four conformations showed clashes between the p53 core domain and the inter-protomer C-lobe (Supplementary Fig. 7b). Therefore, we used the complex model of the E6AP/E6/p53 core domain, generated by docking p53 onto the E6AP/E6 structure of Det1, to investigate how E6AP/E6 protomeric dimerization contributes to p53 ubiquitination.

Among the residues within the p53 core, the C-terminal residues K291 and K292—both ubiquitination sites[39]—reside in proximity to the catalytic C843 of the E6AP C-lobe of another protomer; indeed, K292

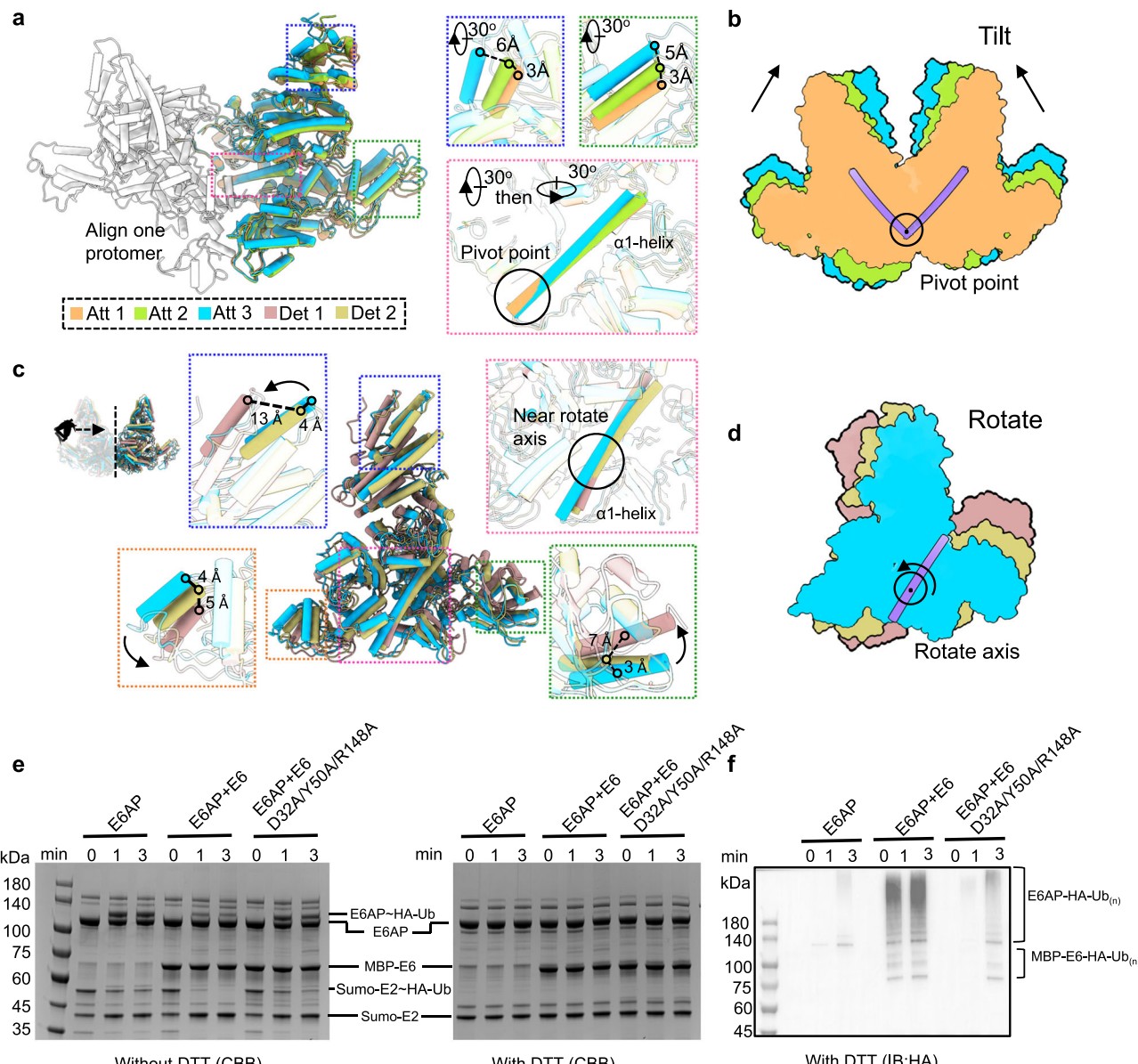

**Fig. 3 | Dynamics of the E6AP/E6 complex. a** Structural comparison of the E6AP/E6 complexes among Att1 (orange), Att2 (green), and Att3 (cyan). One (gray) of the two E6AP/E6 protomers is superimposed. Insets are zoomed-in views of the boxed regions in **a**, showing structural differences among the three states. The region of the α1-helix showing no conformational change is indicated by a circle. **b** Schematic illustration of the structural comparisons of Att1-3, showing conformational changes in a tilting fashion from Att1, through Att2, to Att3 using the intersect point of the two α1-helices as a pivot point. **c** Structural comparison of the E6AP/E6 complexes among Att3, Det1 (gold), and Det2 (pink). Insets are zoomed-in views of the boxed regions in **c**, showing the structural differences among the three states. The region of the α1-helix showing minimum conformational change is indicated by a circle. **d** Schematic illustration of the structural comparison of Att3, Det1, and Det2, showing conformational adaptation in a rotation fashion from Att3, through Det2, to Det1 using a region at the α1-helix as a rotation axis. **e** E2 (UbcH7)-E3 (E6AP) transthiolation assay. E2, E6, and ubiquitin (Ub) have the N-terminal Sumo, MBP, and HA tags, respectively. The symbol (-) means that ubiquitin is attached to the active site of E6AP or E2 via a reducing agents-sensitive thioester bond such as DTT. **f** Immunoblotting analysis for the reaction products in the right panel of **e**. The symbol (−) means that ubiquitin is attached to the modified site via a reducing agents-insensitive to isopeptide bond. $Ub_{(n)}$ refers to the polyubiquitin chain of "n" length. The efficiency with which E6AP transfers ubiquitin to the modified sites of substrates ($E6AP-Ub_{(n)}$ or $E6-Ub_{(n)}$) is positively correlated with the reduction rate in E6AP-Ub. Source data are provided as a Source Data file.

of one protomer is about 31 Å away from the catalytic C843 of the C-lobe in the other protomer (Supplementary Fig. 7a). A previous study showed that K305 of p53 is the major ubiquitination site employed by the E6AP/E6 complex[39]. We presume that the 13-residue extension from K292 to K305 renders K305 more accessible to C843 than K291 or K292 (Supplementary Fig. 7a). Meanwhile, as observed in the E6AP/E6 complex, the two heterotrimers should orientate towards each other, creating proximity between the p53 core and the C-lobe until contact is made, promoting the aberrant ubiquitination of p53.

Intriguingly, as compared to the inter-protomer distance, the intra-protomer distance between C843 in the C-lobe and K292 of p53 core is obviously larger at ~66 Å (Supplementary Fig. 7a). Furthermore, the C-lobe and the p53 core within one E6AP/E6/p53 core protomer are separated by the NTR (Supplementary Fig. 7a), rendering them inaccessible to each other. To verify that p53 would not be ubiquitinated by the intra-protomeric E6AP, we constructed a mutant R505P of E6AP, a known loss-of-function mutation associated with Angelman syndrome. The mutation is located at the α1-helix that mediates the dimerization

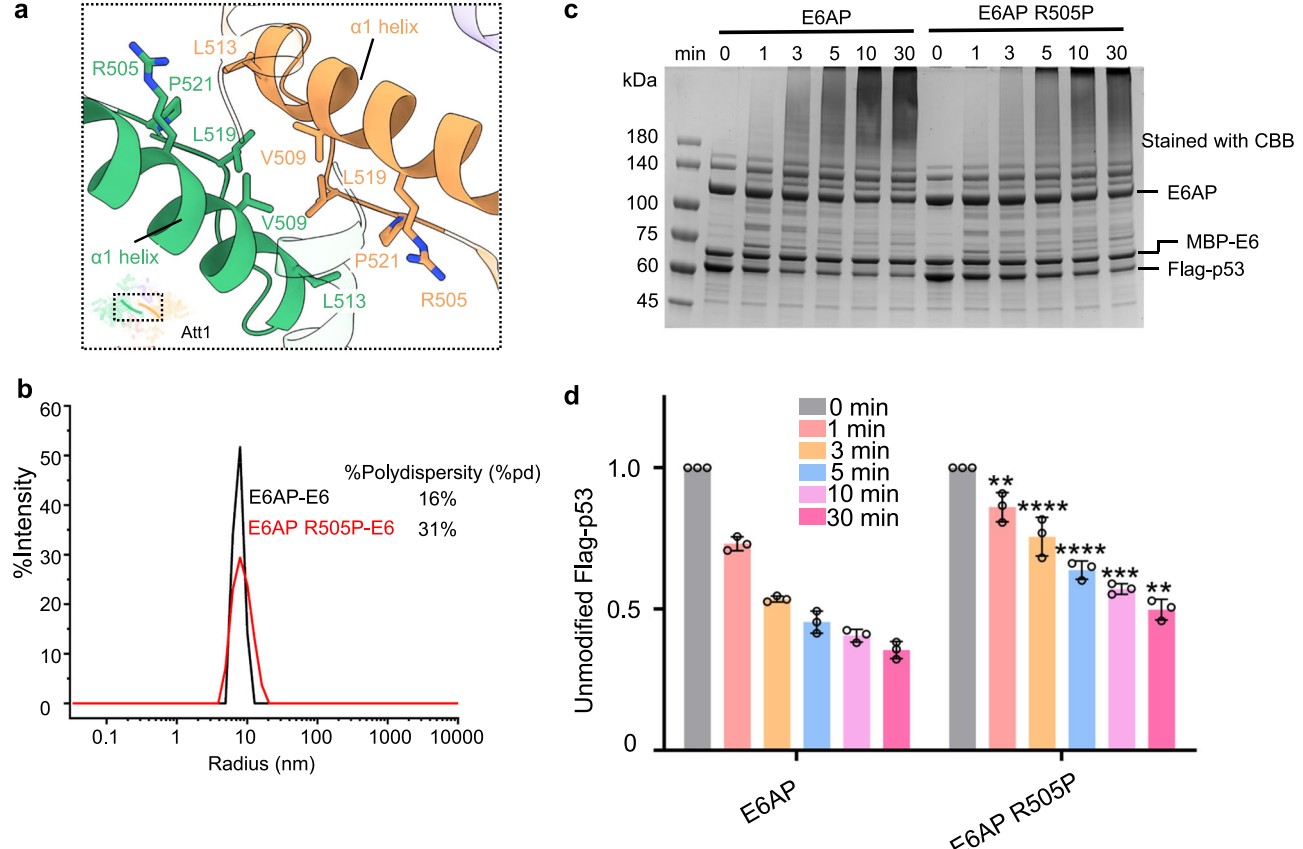

**Fig. 4 | An Angelman syndrome-related mutation decreases the E6-stimulated activity of E6AP. a** Cartoon showing the location of the Angelman syndrome-related mutation, R505P. The small cartoon in the lower left corner is shown for orientation purposes, and indicates a close-up view enclosed by the black line. E6AP appears green in protomer 1, whereas in protomer 2, it is yellow. **b** Effect of E6AP R505P mutation on the stabilization of the E6AP/E6 complex using the dynamic light scattering (DLS) assay. %Polydispersity (%pd) is a measure of sample stability, with larger %pd indicating higher instability. **c** Effect of the E6AP R505P mutation on the ubiquitination of the substrate p53. The activity of E6AP toward p53 is negatively correlated with the amount of unmodified p53. **d** Statistics of the enzymatic activity in **c**. Statistical significance tests compare the amount of unmodified p53 by the E6AP R505P mutant with that by wild-type E6AP at the same time point. Data are presented as the mean ± SD of triplicate experiments. ****p < 0.0001 and ***p < 0.001 based on one-way analysis of variance (ANOVA) with Tukey's multiple comparison test. Source data are provided as a Source Data file.

of two protomers (Fig. 4a). The dynamic light scattering (DLS) assay [40,41] shows that the polydispersity (pd) of the mutant is significantly higher than that of the wild type, indicating that the mutant E6AP/E6 complex has a wider particle size distribution [40,41], therefore is much less stable than the wildtype E6AP /E6 complex (Fig. 4b). The mutation also significantly reduced the level of p53 ubiquitination (Fig. 4c, d and Supplementary Fig. 8a, b). Collectively, these finding suggest that dimerization of the E6AP/E6 protomer is necessary for efficient p53 ubiquitination.

### The intra-protomer interaction between E6 and E6AP
E6 binds to E6AP through two binding pocket interactions (Fig. 5a and Supplementary Fig. 9a). In the first, a short fragment containing the LxxLL motif from the loop-helix-loop element of E6AP inserts into a cleft formed by E6N and E6C (Supplementary Fig. 9a), which is similar to that presented in previously reported structures (Supplementary Fig. 9b) [27,29]. In the second, the E6C domain and the E6 linker helix insert into a pocket formed by various elements: the α1-helix of the N-lobe, two short helices of the tower, and the loop-helix-loop element (Fig. 5a).

The binding pocket of E6AP accommodates E6 mainly through hydrophobic and polar interactions (Fig. 5b). At the upper edge of the pocket opening, there are three types of interactions (Fig. 5c). The first is β-augmentation, wherein a three-stranded β-sheet from E6C is augmented by a short β-strand (residues 468-470) from E6AP's tower region. The second is an anion···ring–cation synthon [42], wherein the Y91 from E6C is sandwiched by R131 of E6C and E472 from the tower region. The third is a cation-pi interaction between R424 of E6AP and Y88 of E6. At the lower edge of the pocket opening, two hydrophobic surfaces are located on the two opposite sides of the E6 linker helix, and these surfaces are sandwiched by two hydrophobic clusters. While the surfaces of L69 and L74 contact a hydrophobic cluster of residues L409, L412, and L413 from the loop-helix-loop element, the surfaces of F76, Y77, I80, and Y83 interact with the hydrophobic cluster of I507, L510, Y511, and V514 from the α1-helix (Fig. 5d). At the bottom of the pocket, three basic residues from E6C—R84, H85 and R136—form a positively charged patch that is surrounded by negatively charged residues, including E504, E403, E399, and E743 from E6AP (Fig. 5e).

To explore the role of the interaction between E6 and the pocket of E6AP, we constructed E6 mutants with a double mutation, Y88A/Y91A, or a triple mutation, F76A/I80A/Y83A, and performed ubiquitination assays. We found that both mutants significantly compromised the activity of E6AP toward its substrate, p53 (Fig. 5f, g), indicating that the pocket is a site where E6 regulates the ligase activity of E6AP.

### Structural changes of E6AP caused by E6 binding
All known structures of HECT-type E3s show a conserved position of the α1-helix with respect to the rest of the HECT domain (Supplementary Fig. 10). As mentioned, the α1-helix of E6AP plays a key role in

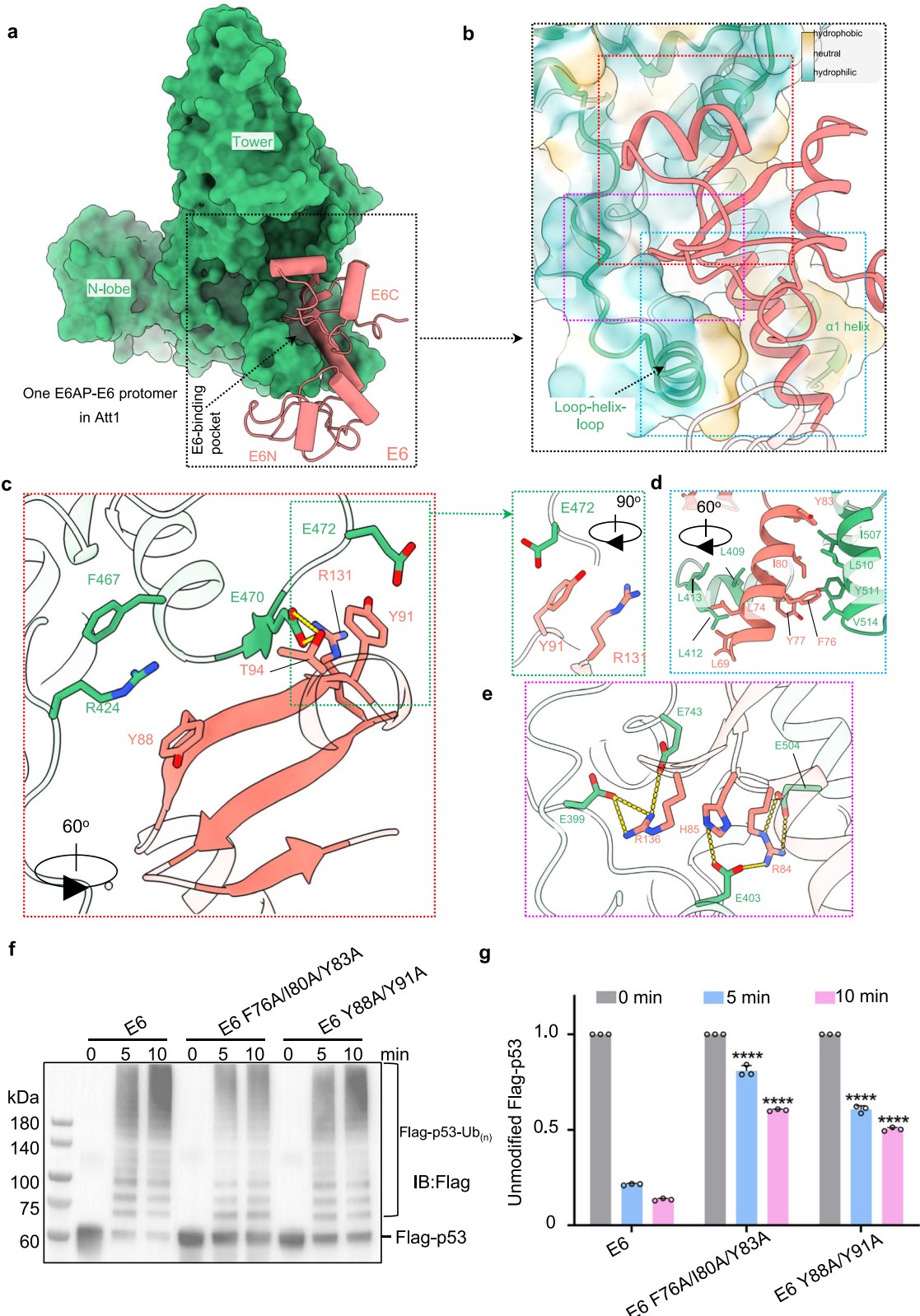

**Fig. 5 | An E6-binding pocket in E6AP. a** E6 (cartoon, red) binds into a pocket in E6AP (surface, green). **b** Zoomed-in view of the interaction between E6 and the E6-binding pocket in E6AP. The E6-binding pocket, depicted as a lipophilicity potential surface (hydrophobic, gold; hydrophilic, cyan), is formed by the α1-helix of the N-lobe, the tower, and the loop-helix-loop element. **c–e** Detailed interactions in the corresponding area outlined by the colored boxes in **b**. **f** Effect of E6AP mutations on ubiquitination of the substrate p53. The mutations are at the E6/E6-binding pocket binding interface. The activity of E6AP toward p53 is negatively correlated with the amount of unmodified p53. **g** Statistics of the enzymatic activity in **f**. Statistical significance tests were performed to compare the amount of unmodified p53 by the wildtype and mutant E6AP at each time point. Data are presented as the mean ± SD of triplicate experiments. ****p < 0.0001 based on one-way analysis of variance (ANOVA) with Tukey's multiple comparison test. Source data are provided as a Source Data file.

mediating the dimerization of the E6AP/E6 protomer. To explore whether and/or how E6AP forms a dimer of itself, we prepared a full-length E6AP sample and performed cryoEM analysis (Supplementary Fig. 11a–h). 2D averaging and 3D classifications showed that, in the absence of E6, E6AP mainly exists as monomer. We identified two 3D classes with discernible and similar overall structures of the NTR, C-lobe and N-lobe domains (Supplementary Fig. 11d, e); albeit, one class with a higher resolution showed extra densities extending from the C-lobe of E6AP, which we attributed to the ubiquitin accepted by the C-lobe of E6AP. The C-lobe, together with these extra densities, fits well with the known structure of the ubiquitin-linked C-lobe (PDB ID: 6FYH[43]) (Supplementary Fig. 11f). Thus, we selected the 3D class showing ubiquitin densities for further data processing and obtained a reconstruction of E6AP at a resolution of 5.1 Å (Supplementary Fig. 11g, h). Structural comparison of E6AP alone and in complex with E6 revealed two major differences: the first was that the loop-helix-loop element (residues 398–420), which is well-resolved in the E6AP/E6 complex, is not observed in the structure of E6AP alone (Fig. 6a); the second was that the α1-helix (residues 486–514) of E6AP from the E6AP/E6 complex is longer than that (residues 486–502) of E6AP in the absence of E6 (Fig. 6b). Since both the loop-helix-loop element and the α1-helix make strong interactions with the E6 (as revealed in the E6AP/E6 complex structures), it is conceivable that E6 binding to E6AP induces and/or stabilizes the extension of the α1-helix of E6AP and the formation of the loop-helix-loop element. Intriguingly, we observed that two of the 2D averages are reminiscent of the dimerized E6AP (Supplementary Fig. 12), which supports previous observations that, in the absence of E6, the auto-ubiquitination of E6AP occurs predominantly by oligomeric form[44,45].

To investigate how E6 binding causes the conformational changes in E6AP, we performed three 500-ns molecular dynamics (MD) simulations: the dimer of the E6AP/E6 protomer, one E6AP/E6 protomer, and the isolated E6AP extracted from the E6AP/E6 complex. The root-mean-square fluctuation (RMSF) profile and snapshots extracted from the simulation trajectories of E6AP in the three simulations showed variable flexibilities among the different systems for residues 398-418 of the loop-helix-loop element and C-terminal residues 503-514 of the extended α1-helix (Fig. 6c, d). The loop-helix-loop element in the isolated E6AP system is more flexible than that of the other two systems, suggesting that this region is stabilized by E6 binding. Comparatively, the C-terminal stretch of residues 503-514 of the α1-helix in the isolated E6AP shows the largest fluctuation among the three systems but the highest stability in the dimer of the E6AP/E6 protomer (Fig. 6c, d).

Evolution of the secondary structure of the α1-helix along the simulation trajectories was monitored by the DSSP[46] program (Supplementary Fig. 13a). The secondary structure transformation profile indicated that the C-terminal stretch of residues 503–514 of the extended α1-helix in the isolated E6AP system undergoes dramatic conformational changes, whereas that in the dimer of the E6AP/E6 protomer system appeared stable. As for the E6AP/E6 protomer system, the C-terminal residues 503-510 maintained helicity during most of the stimulation, whereas the 510–514 stretch of the extended α1-helix tended to unfold. Furthermore, when we carried out the 500-ns MD simulation for the isolated E6AP dimer (without E6, as extracted from the E6AP/E6 complex), we found the extended parts of the α1-helices to be unfolded (Supplementary Fig. 13b, c). Put together, our MD simulations show that E6 binding is essential for the stabilization of the extended α1-helix.

## Discussion

For substrate ubiquitination, the C-lobe of the HECT-type E3 needs to accept ubiquitin from an E2 and donate it to its substrates[7]. In this study, we solved the structures of the E6AP/E6 complex in five different states, including three Att states and two Det states. The Det and Att conformations of the E6AP/E6 complex presented in this study should be structurally relevant to these accepting and donating processes, respectively. Unlike the N-lobe of E6AP, which is restricted by strong interactions with its N-terminal region, the C-lobe in the Det conformation is free to move around and accept ubiquitin from an E2; this is evident in the overlay structure of the HECT/E2–ubiquitin complex presented in Supplementary Fig. 14a. Once loaded with ubiquitin, the C-lobe of E6AP sways toward E6 and/or its bound p53 until the two make contact, and the conformation of the E6AP/E6 complex changes from Det to Att (Supplementary Figs. 7a, 14b and Fig. 6e). Intriguingly, as either E6 or p53 is found to be polyubiquitinated by E6AP[38,39], the conformational changes to the E6AP/E6 complex should be cyclical not unidirectional: after donating ubiquitin, the unloaded C-lobe would detach and move away from the substrate to accept ubiquitin again from the E2 for further ubiquitination, leading to polyubiquitination of E6 and/or p53.

Previous studies have shown that E6 can enhance E6AP activity through increased oligomerization[44,47]. Our structural and functional studies reveal how the E6 binding transforms the inactive E6AP monomer into an active dimer by inducing and/or stabilizing the extension of the α1-helix (Fig. 6e). Given that a small portion of E6AP particles were observed in a dimeric form (Supplementary Fig. 12), it is possible that the short α1-helix of E6AP, under normal physiological conditions, can transiently transform into the extended state, with only the extended state capable of mediating E6AP dimerization and function.

Intriguingly, some loss-of-function (R500P, M501I, and R505P) and gain-of-function (T508A) mutations in E6AP[35–37], which are reported to contribute to Angelman syndrome and Autism Spectrum Disorder, occur on the extended α1-helix (Fig. 4 and Supplementary Fig. 15a). The effects of these mutations are correlated with the stability or interactability of the extended α1 helix, highlighting the importance of the α1-helix in mediating dimerization interactions.

Indeed, the three loss-of-function mutations (R500P, M501I, and R505P) would reduce the stability of the extended α1-helix, presumably because they are involved in maintaining the structure: residue R500 forms a salt bridge with D496 (Supplementary Fig. 15b) and residues M501 and R505 form hydrogen bonds with Q554 (Supplementary Fig. 15b). The mutations of R500P, M501I and R505P would abolish these interactions. Furthermore, the proline substitution mutations at residues R500 and R505 would also create a kink in the helix[48,49], and likely disrupt its extension. Therefore, the three mutations compromise the E6AP activity likely by undermining the stability of the extended α1-helix, and thereby decrease the dimerization of E6AP. Consistently, we show that, one of the aforementioned E6AP mutations, R500P, destabilizes the E6AP/E6 complex (Fig. 4b) and significantly reduces p53 ubiquitination (Fig. 4c, d).

On the other hand, a T508A gain-of-function mutation, associated with Autism Spectrum Disorder, would presumably enhance the interaction between the two extended α1-helices. Residue T508 is immediately adjacent to the hydrophobic interface of the extended α1-helix, as can be seen best in Supplementary Fig. 15c. A mutation at this residue would obviously increase the hydrophobic interface of the extended α1-helix and facilitate an interaction between the two extended α1-helices. Furthermore, E6AP activity is down-regulated by the phosphorylation of T508[35]. The high hydrophilicity of the phosphate group would drastically decrease the hydrophobic interaction of the extended α1-helicies. Consistently, a previous study has shown that E6AP bearing an T508A mutation engages in strong interactions with itself, whereas a phospho-mimic mutant T508E displays little self-interaction[35]. The enhancing interaction between the two extended α1-helices would likely increase the dimerization potential of E6AP, thus causing its higher activity than the wild type. These ideas support the notion that the α1-helix–mediated conformational transition of E6AP from a monomer to a dimer may also control its activity under physiological conditions.

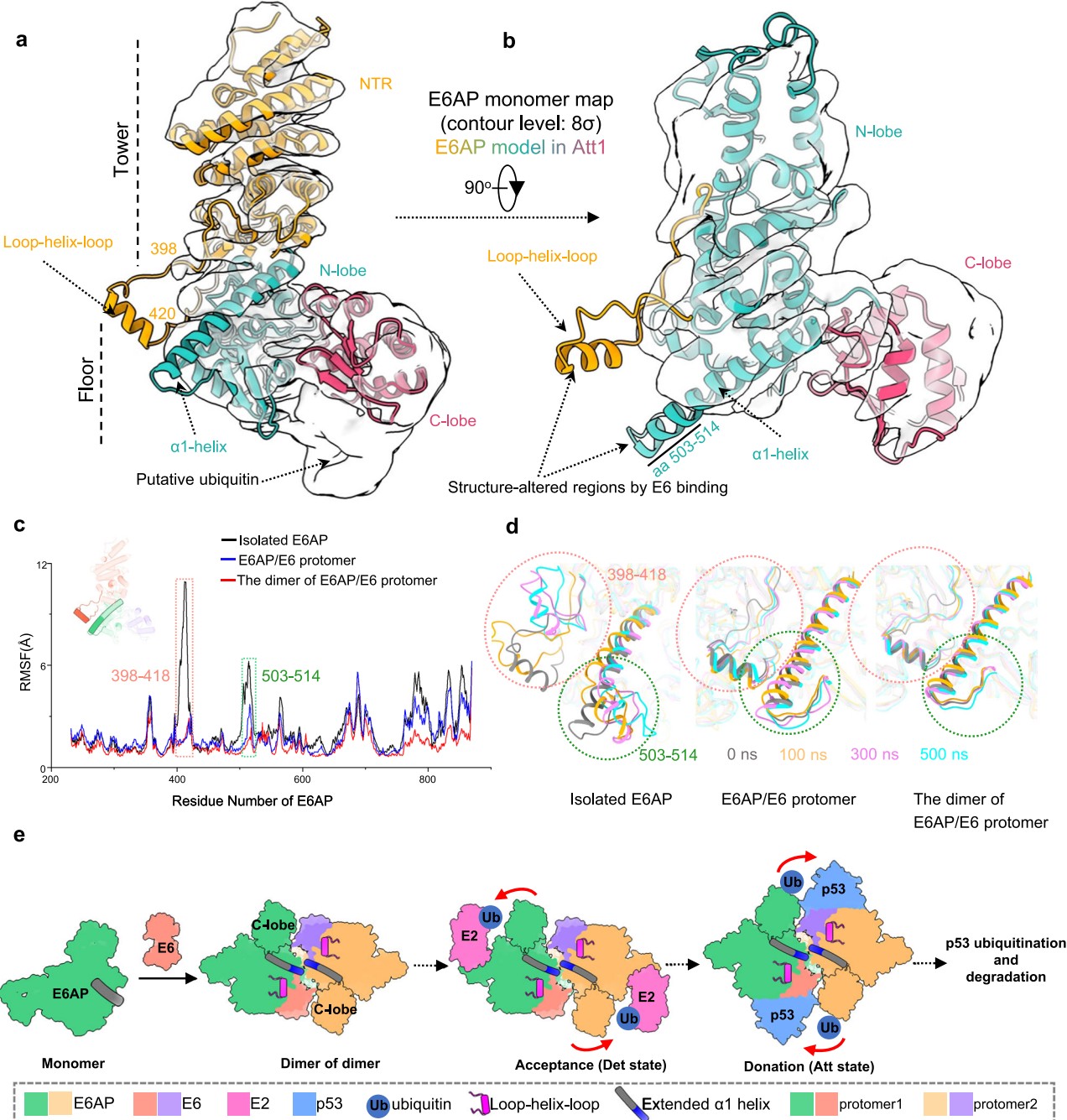

**Fig. 6 | Structural rearrangement of E6AP upon binding with E6. a**, **b** Side (**a**) and top (**b**) views of the docking of the E6AP model from E6AP/E6 complex in Att1 into the density map of the E6AP monomer. N-terminal region (NTR), N-lobe and C-lobe of E6AP are yellow, green, and red, respectively. **c** RMSF profiles for each residue of E6AP over the 500-ns simulation for each of the three systems containing the dimer of the E6AP/E6 protomer (red), one E6AP/E6 protomer (blue), and the isolated E6AP (as extracted from the E6AP/E6 complex, black). Flexible regions are highlighted, which are inside the loop-helix-loop element (residues 398-418, red box) and the α1-helix (residues 503-514, green box). **d** Alignment of the E6AP snapshot structures during the 500-ns simulation for the three systems. The initial structure (0 ns) and the snapshot structures at 100, 300, and 500 ns are colored in gray, yellow, red, and blue, respectively. **e** A working model showing the hijacking activity of the E6AP ubiquitin ligase by HPV E6 to ubiquitinate p53. The red arrows indicate the proximity of the C-lobe of E6AP to E2 or to substrates when two E6AP/E6 protomers sway around the crossed region of the two extended α1-helices. The short α1-helix of the E6AP monomer is shown in grey. The extended parts of the long α1-helix in the E6AP/E6 are shown in blue. Source data are provided as a Source Data file.

## Methods

### Constructs

Human E6AP (residues 1-875, uniprot Q05086-1), Human p53 (residues 1-393, uniprot P04637), and HPV16 E6 (residues 1-158, uniprot P03126) were cloned into pFastBac vectors with an N-terminal strep-, Flag-, and His6 tag, respectively, using homologous recombination (CloneExpress One Step Cloning Kit, Vazyme). For the prokaryotic expression,

HPV16 E6 was cloned into a modified version of the pMAL-c2× vector, with the resultant protein containing an MBP tag followed by a tobacco etch virus (TEV) protease-cutting site in its N-terminus. Mutant E6 4 C/ S, bearing four mutations at non-conserved cysteine residues, not only prevents disulfide-mediated aggregation but retains a potency for E6AP-mediated p53 ubiquitination that is comparable with the wild type[27,29]. This mutant was used for structural determination and

biochemical assays. All mutations used in this study were created using a standard PCR-based mutagenesis method and confirmed by DNA sequencing. All primer sequences used in this study are listed in Supplementary Table 2.

## Protein expression and purification

For constructs in the pFastBac vectors, recombinant proteins were expressed in Sf9 insect cells (Invitrogen, 11496015) using the Bac-to-Bac baculovirus expression system (ThermoFisher). For constructs in the pMAL-c2x vector, recombinant proteins were expressed in *Escherichia coli* BL21 (DE3) host cells (New England Biolabs). Cell pellets were collected and resuspended in a lysis buffer containing 20 mM HEPES pH 7.4, 150 mM NaCl, and a protease inhibitor cocktail (Bimake). Cells were lysed by high-pressure homogenization and centrifuged at $100,000 \times g$ for 30 min at 4 °C to remove cell debris. The supernatant was incubated with Flag (Genscript), His6 (SMART Lifesciences), Strep (IBA-Lifesciences) or MBP (SMART Lifesciences) affinity resins, depending on the tags. The captured proteins were washed thoroughly and eluted using the same lysis buffer supplemented with 200 μL/mL Flag peptide, 250 mM imidazole, 50 mM biotin, or 10 mM maltose, respectively. The eluted proteins were subjected to size-exclusion chromatography (SEC) on a Superdex 6 Increase 10/300 column (GE Healthcare) equilibrated with a buffer containing 50 mM Tris (pH 8.0) and 100 mM NaCl. Fractions from the SEC were evaluated by sodium dodecyl sulfate-polyacrylamide gel electrophoresis (SDS-PAGE) and those fractions containing the target protein were pooled and concentrated for use.

E6 can increase the auto-ubiquitination of E6AP[44,47,50]. To assemble the E6AP/E6 complex while preventing E6 aggregation and the auto-ubiquitination of E6AP, we co-expressed the mutant E6AP C843A with the mutant E6 4 C/S in Sf9 insect cells with N-terminal strep and His6 tags, respectively. After two-step tandem affinity purification and SEC purification, we obtained a complex with good purity and homogeneity. For the preparation of the cryo-EM sample of E6AP alone, we expressed the wild-type E6AP with an N-terminal strep tag in Sf9 insect cells and then purified the sample using strep affinity purification and SEC.

## Cryo-EM grid preparation and data acquisition

To prepare cryo-EM grids, 2.5 μL of the purified protein at 0.5 mg/mL for E6AP or 4 mg/mL for the E6AP/E6 complex were applied, respectively, onto glow-discharged Quantifoil R1.2/1.3 300-mesh gold holey carbon grids. The grids were blotted under 100% humidity at 4 °C and then plunge-frozen in liquid ethane cooled by liquid nitrogen using a Vitrobot Mark IV (Thermo Fisher Scientific). The frozen grids were stored in liquid nitrogen for data acquisition.

Movies were collected on a 300 kV Titan Krios (FEI) equipped with a Gatan K3 Summit direct electron detector and a Gatan energy filter operated with a slit width of 10 eV (GIF). SerialEM[51] was used to automatically acquire micrographs at a pixel size of 1.071 Å and defocus values ranging from -1.5 to -2 μm. A total of 3,536 movies for E6AP were collected at a total dose of 72 e-/Å$^2$ over an exposure time of 3.4 s with a total of 40 frames per movie. A total of 21,152 movies for the E6AP/E6 complex were collected at a total dose of 50 e-/Å$^2$ over an exposure time of 2.1 s with a total of 56 frames per movie.

## Image processing and 3D reconstruction

All frames in each movie were aligned and dose-weighted using MotionCor2[52]. Gctf was used to estimate defocus values and astigmatism parameters of the contrast transfer function (CTF)[53].

For the sample of the E6AP/E6 complex, 20,503 good micrographs were selected after discarding those with crystal ice contamination. About 5,000 particles were manually picked in RELION 3.0[54] from selected micrographs with good contrast. The particles were extracted and subjected to a round of reference-free 2D

classification. 2D averages with different orientations were selected and used as templates for automatic particle picking. A total of 16,234,980 particles are extracted and processed with multiple rounds of 2D classifications, yielding 1,113,449 particles generated from 2D averages with high-resolution structural features. An initial model for 3D classifications was generated de novo from the selected particles using the stochastic gradient descent (SGD) algorithm. Two-fold symmetric features were observed both in 2D averages and in the reconstructed map of E6AP/E6 complex. We thus used C2 symmetry thereafter for 3D classification and map refinement. After rounds of 3D classification, one of the four generated maps displayed a detached interface between the E6AP C-lobe and E6, while the remaining three maps had an attached configuration at the corresponding region.

For structure determination of the E6AP/E6 complex in its various attached forms, we selected particles from two of the three attached forms with high-resolution features and subsequently performed 3D refinement, CTF refinement, and Bayesian polishing of particles. Following this, we obtained the density map; albeit this had poor densities at the upper region of the E6AP tower. Therefore, we performed an additional round of focused alignment with a small mask encompassing only the upper region of E6AP tower and set the Tau value to 80. In doing so, we were then able to sort particles into classes with different conformations for the upper region of the E6AP tower. This resulted in three maps with different distances between the two E6AP towers, containing 233,242 (Attached form 1, Att1), 144,967 (Att2), and 41,041 particles (Att3). Following this, we separately reconstructed the half maps for these three classes with final resolutions of the three reconstructions determined at 2.6 Å (Att1), 3.1 Å (Att2) and 3.6 Å (Att3).

For structure determination of the E6AP/E6 complex in its detached forms, 3D classification was first used to remove bad particles, followed by a round of 3D refinement with particles from the detached map. This generated a density map with a relatively low resolution. We performed a round of focused alignment with an E6AP/E6 mask and obtained two converged maps with different rotated 2-fold interfaces that contained 22,398 (Det1) and 33,787 particles (Det2). We then reconstructed the half maps for the two classes, and the final resolutions of the two reconstructions were determined at 4.2 Å (Det1) and 4.4 Å (Det2).

For the monomeric E6AP protein, manual particle picking in RELION 3.0[54] from 3,536 selected micrographs yielded 8,000 particles. These particles were extracted and subjected to reference-free 2D classification. The selected 2D classifications were used as templates for automated particle picking. A total of 5,011,601 particles were picked and processed with several rounds of 2D classification to discard any poorly defined particles. The selected particles representing different orientations of E6AP were subjected to rounds of 3D classifications. One of the two types of converged classes contained extra densities at the bottom that fitted well with the structure of ubiquitin. Aside from this extra density, the two types of 3D averages had a similar overall structure. Thus, the particle set from the ubiquitin-containing reconstruction that had higher resolution structural features was extracted and subjected to the 3D refinement, yielding a reconstruction of E6AP at a global resolution of 5.1 Å.

The global resolution was estimated using the gold-standard Fourier shell correlation (FSC) at a threshold of 0.143. Local resolution was calculated by ResMap[55]. To improve the quality of the maps, two half maps from the final refinement were provided without a mask as input to DeepEMhancer[56].

## Model building and refinement

To build the atomic models of the E6AP/E6 complex in the Att1, Att2, and Att3 states, the AlphaFold[57,58] model of E6AP and the previous reported structures of E6 (PDB ID: 4XR8)[29] were docked onto the density maps of Att1, Att2 and Att3, respectively, using Chimera[59] and then manually adjusted in Coot[60]. Because we are unable to determine

the sequence for the N-terminal fragments at the reconstruction center based on the density map of the E6AP/E6 complex, we only traced and built two symmetric Cα models of 14-residue-long helix-loop motifs in that region. The different models were combined and refined respectively against the corresponding density maps using Phenix[61].

To model the E6AP/E6 complex in the Det1 and Det2 states—owing to the moderate resolution of their density maps—we fitted the complex model from Att1 states into the two maps and manually adjusted the Cα models in Coot to remove the disordered regions. Finally, the models of the E6AP/E6 complexes in the Det1 and Det2 states were refined against the corresponding density maps using Phenix. The figures were prepared using Chimera and ChimeraX[62].

### In vitro ubiquitination assay

For the substrate ubiquitination assay, the purified Flag-p53, with the wild-type or the mutant E6, was incubated with 60 nM E1 (UBE1), 400 nM E2 (UbcH7), 400 nM E6AP, and 30 μM ubiquitin in ubiquitination reaction buffer. The reactions at 37 °C were initiated by the addition of ATP and quenched by mixing the reaction mixture with protein loading dye (Sangon Biotech) at the indicated time points. Samples were resolved by sodium dodecyl sulfate-polyacrylamide gel electrophoresis (SDS-PAGE) and subjected to immunoblotting (IB) according to standard laboratory procedures, as described below. The unmodified p53 bands at the distinct time points shown were quantified and normalized to the zero time points.

### Ubiquitin transfer assay

The ubiquitin transfer assay was performed to monitor ubiquitin transfer from E2 (UbcH7) to E6AP according to a previously described method[63–65], with some modifications. In brief, 5 μM UbcH7 was reacted with 0.25 μM E1 and 10 μM ubiquitin in a buffer containing 50 mM Tris (pH 7.6), 250 mM NaCl, 5 mM MgCl$_2$ and 5 mM ATP for 30 min at 37 °C. Reactions were quenched by diluting the samples four-fold with a solution of 25 mM HEPES (pH 7.5), 100 mM NaCl, and 25 mM EDTA. Subsequently, the pre-charged E2-Ub ("~" represents the thioester bond, which is sensitive to the reducing agent) mixture and excess E6AP were mixed without or with MBP-E6 to initiate a single round of substrate ubiquitination. Reactions were quenched with an equal volume of protein loading dye without or with reducing agent at the indicated timepoint(s). Reaction products were resolved with SDS-PAGE and detected using Coomassie brilliant blue (CBB) or IB.

### Immunoblotting

Proteins were transferred to a 0.45-μm nitrocellulose membrane (Millipore, IPVH00010) and then blocked with 3% bovine serum albumin in Tris-buffered saline with Tween-20 (TBST, 20 mM Tris-HCl, pH 7.4, 137 mM NaCl and 0.1% Tween 20) at room temperature for 1 h. Membranes were then incubated with anti-Flag antibody (ABclonal, AE005, 1:3000) or anti-HA antibody (ABclonal, AE008, 1:3000) at 4 °C overnight. The membranes were washed three times with TBST buffer, incubated with horseradish peroxidase-conjugated goat anti-mouse antibody (ABclonal, AS003, 1:4000), and visualized on a ChemoDoc MP imaging system (BIO-RAD).

### Static light-scattering (SLS) analysis

The purified E6AP/E6 complex (100 μl at ~3 mg/mL) was injected into an AKTA fast protein liquid chromatography system with a SRT SEC-300 column (Sepax Technologies) and a column buffer of 20 mM Tris (pH 8.0) and 100 mM NaCl. The chromatography system was coupled to an SLS detector (miniDawn; Wyatt) and a differential refractive index detector (Optilab; Wyatt). Data were analyzed with ASTRA 6 (Wyatt).

### Dynamic light-scattering (DLS) analysis

E6AP and equimolar MBP-E6 were incubated at 4 °C. After 1 h, the mixture was subjected to dynamic light-scattering (DLS) analysis using Wyatt DynaPro PlateReader III. Particle size was determined from an average of five correlation functions and analyzed using Dynamics V7. Polydispersity (pd) is a measure of sample stability, with a larger pd indicative of higher instability.

### Molecular dynamics (MD) simulations

The systems of the isolated E6AP, one E6AP/E6 protomer, and the E6AP dimer were constructed by extracting one EA6P monomer, one EA6P/E6 protomer, or one E6AP dimer from a dimer of the E6AP/E6 protomer. The four systems (the dimer of E6AP/E6 protomer, one E6AP/E6 protomer, the isolated E6AP, and E6AP dimer) were refined by the Protein Preparation Wizard Workflow integrated in Maestro (Version 9.0; Schrödinger, LLC) and all parameters were set to the default settings. The protonation states of all titratable residues were determined using PROPKA[66] at pH 7.0.

MD simulations were carried out using Gromacs 2019.6 program package[67] with Amber ff99sb*-ILDNP force-field[68]. Periodic boundary conditions were used to avoid edge effects in all calculations. Each system was solvated in a cubic box with TIP3P water molecules to keep the boundary of the box at least 10 Å away from the protein. Na$^+$ ions were subsequently added for charge neutralization. Minimization and equilibration were performed for all calculations. First, 1000 steps of the steepest descent energy minimization were applied. Then, the minimized systems were heated from 0 to 300 K by NVT MD simulations. Afterwards, a multistage NPT equilibration protocol was applied by gradually decreasing the positional restraints on all backbone and Cα atoms of the proteins. Finally, all restraints were removed for the production run. The results of the MD simulations were analyzed using the Gromacs tools.

### Statistics and reproducibility

All assays were repeated at least three times and showed comparable results. Statistical parameters including the definitions and exact values of n (number of experiment) are reported in the corresponding figure legends. All data are presented as mean ± SD of triplicate experiments; ns, not significant, $*p < 0.05$, $**p < 0.01$, $***p < 0.001$, and $****p < 0.0001$ using one-way analysis of variance with Tukey's multiple comparison test. All statistical data were performed in GraphPad Prism 6.

### Reporting summary

Further information on research design is available in the Nature Portfolio Reporting Summary linked to this article.

## Data availability

All density maps generated in this study have been deposited in the Electron Microscopy Bank under accession codes EMD-36604 (the monomeric E6AP with a putative ubiquitin), EMD-36599 (the E6AP/E6 complex in the Att1), EMD-36600 (the E6AP/E6 complex in the Att2), EMD-36601 (the E6AP/E6 complex in the Att3), EMD-36602 (the E6AP/E6 complex in the Det1), and EMD-36603 (the E6AP/E6 complex in the Det2). The atomic coordinates generated in this study have been deposited in the Protein Data Bank under accession code 8JRN (the E6AP/E6 complex in the Att1), 8JRO (the E6AP/E6 complex in the Att2), 8JRP (the E6AP/E6 complex in the Att3), 8JRQ (the E6AP/E6 complex in the Det1), and 8JRR (the E6AP/E6 complex in the Det2). Source data are provided in this paper. The previously-published atomic coordinates used in this study have been deposited in the Protein Data Bank under accession codes 7MWD, 7BII, 6J1X, 1ZVD, 5TJ8, 5XMC, 2XBF, 3JW0, 5HPL. Source data are provided as a Source Data file. Source data are provided in this paper.

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

## Acknowledgements

The cryo-EM data were collected at the Cryo-Electron Microscopy Research Center, Shanghai Institute of Material Medica. This work was partially supported by the National Key Research and Development Program of China (2022YFC3400500 to X.Y., 2021ZD0203900 to C.L. and 2022YFC2804800 to Z.L.); the Hundred Talents Program of the Chinese Academy of Sciences (to X.Y.); the Lingang Laboratory (LG202103-02-08); the Shanghai Municipal Science and Technology Major Project (TZX022021007); the National Natural Science Foundation of China (32000896 to Z.W., 81821005 and 92253303 to C.L.); the High-level New R&D Institute (2019B090904008 to C.L.), the High-level Innovative Research Institute (2021B0909050003 to C.L.); the Department of Science and Technology of Guangdong Province (to C.L.); support from the SANOFI Scholarship Program (to Z.W.); the Zhejiang Provincial Natural Science Foundation of China (LY23H190002 to Z.L.); and Research Funds from Hangzhou Institute for Advanced Study (2022ZZ01010 to Z.L.); and the Fundamental Research Funds of Zhejiang Sci-Tech University (22042292-Y to F.Y.).

## Author contributions

Z.W. and F.F. designed the constructs, purified the proteins, performed the cryo-EM grid preparation and functional studies, and participated in data collection; J.Q., W.X. and Y.L. performed plasmid construction and participated in functional studies; Q.W. and R.G. performed data collection; F.Y., M.L. and C.L. performed the MD simulation; F.F., X.Y. and Z.L. performed the cryo-EM image processing, structure determination, and model building; X.Y., C.L., Z.W., F.F., Z.L. and F.Y. analyzed the data; Z.L., Z.W., F.F. and F.Y. prepared the figures; X.Y. and Z.W. initiated the project; C.L. supervised F.Y., Z.L., Y.L. and M.L.; X.Y. supervised the project; Z.W. prepared the draft of the manuscript; X.Y., C.L., Z.W., Z.L. and F.Y. wrote the manuscript with input from all authors.

## Competing interests

The authors declare no competing interests.
