## [Peer Review File · Nature Communications]

Reviewers' Comments:

Reviewer #1:

Remarks to the Author:

The work presented by Wang et al. consists of results from structural analyses performed by cryo-EM on the E6AP E3 ubiquitin ligase, both in the free state and bound to the HPV E6 oncoprotein. Whereas free E6AP is essentially observed in a monomeric state, the E6/E6AP complex dimerizes. The resulting E6/E6AP heterotetramer is held in place by inter-protomer contacts involving mainly residues of E6AP within the alpha1-helix but also of the N-helix of E6AP. In the tetramer, the E6 protein from one protomer is in close proximity with the C-lobe of the E6AP HECT domain from the other protomer. Different conformational states of the E6/E6AP tetramer are observed, which are clustered into two main conformational families: the "Attached" and "Detached" conformations. The "Attached" conformations display closer contacts between E6 and the C-lobe as compared to the "Detached" conformations. Such tetrameric organization suggests an inter-protomer mechanism for E6 ubiquitination by E6AP. The authors then used the different E6/E6AP structures to dock the core domain of p53. Only one structure from the Detached family can accommodate p53 without clashes. Based on this model, p53 ubiquitination would also happen through a trans-protomer mechanism but would require a "Detached" conformational state.

In summary, the authors provide very elegant structural data in support of long-standing issue in the field. My main concerns are on the results from biochemical/functional analyses in support of the trans-protomer mechanism.

Major issues:

Fig 2. Panel f: I do not see clear differences in E6 band intensities between wt E6 and E6 D32A/Y50A/R148A. From this experiment it seems to me that suppression of the E6/C-lobe contacts has no effect on E6 ubiquitination. Panel g: the details of this experiment are not described. Which is the protein with the HA tag? I guess it is ubiquitin...How are the authors sure that what they see is E6 ubiquitination and not E2 ubiquitination for example?

Fig5, panel h: here the E6 mutants tested affect residues located at the E6/E6AP interface adjacent to the LxxLL motif region. Results show that these E6 mutants have reduced p53 ubiquitination activities. However, they do not prove that the trans-protomer mechanism is required for p53 ubiquitination. To show this, they would have to make mutants that impair E6AP dimerization through the alpha-1 helix of E6AP and test them for p53 ubiquitination. This is an important experiment. Indeed, one should also keep in mind that full-length p53 is a tetramer and that this could have an impact on the p53 ubiquitination mechanism.

Major issues:

-Please clarify in the text what are the E6 constructs used in the different experiments. Is it wt HPV16 E6?

-Please refer to Sailer C et al Nat Commun 2018. In this paper E6 has already been shown to enhance E6AP activity by increasing oligomerization.

-There are several paragraphs that require rephrasing/corrections. Please revise the text.

Reviewer #2:

Remarks to the Author:

Wang et al. present the structure of the HECT-E3 E6AP in complex with the viral protein E6, revealing a complex that incorporates two copies of each protein. This observation raises pivotal mechanistic questions regarding the implications of such dimerization, which the authors duly explore. Given the recent surge in elucidating HECT ligase structures and insights derived through cryo-EM, the manuscript is particularly timely. Nonetheless, I posit that some of the conclusions drawn are not wholly substantiated by the data presented, and that the structural work may benefit from further refinement. Consequently, I suggest that the authors consider revising their manuscript, taking into account the following concerns and recommendations.

1. I appreciate the authors sharing their maps and models, many of which are of commendable quality. Nevertheless, some issues of overfitting and anisotropy are evident in the map, as

demonstrated through the excessively high-resolution estimates. Att1 certainly does not achieve sub-3Å resolution. A possible explanation might be that the dataset contains a considerable amount of noise. The authors initially selected 16 million particles, utilizing only a small fraction following 2D classification, indicating a significant number of false positives during picking. I recommend exploring alternative picking tools, such as cryoLO or Topaz, to enhance this step, as 2D classification is not effective at removing noisy particles. A reprocessing of the data, potentially employing DeepEMhancer, may further enhance map quality.

2. An issue arises when observing that the maps and models are misaligned when visualized in Chimera or Coot, which could potentially produce inaccurate statistics. The root cause of this misalignment warrants investigation.

3. The model of the monomer map should be restricted to docking and not subjected to building and refinement due to the inadequate quality of the map. Refinement may introduce more errors than it ameliorates.

4. While the paper employs C2 Symmetry, it also explores movements, which appear to lack symmetry, as evidenced by the smeared-out density of the tower. A consideration of C1 processing might provide additional insights.

5. Although the paper posits a sequential order of movements with "towers" modulating the pivot point of the alpha helix, the data does not conclusively support this. Moreover, the utilization of discrete structures on apparent continuous movements is peculiar. Implementing a modern flexibility analysis method, such as 3DVA, 3DFlex, or cryoDRGN, might yield deeper insights into the dynamics.

6. While the authors posit that dimerization controls activity, this is not empirically demonstrated. The reversibility of dimerization is not addressed, nor are the specific E6AP regions required for dimerization experimentally explored. Implementing site-directed mutagenesis and utilizing SLS or EM might provide necessary confirmation of specific interface involvement.

7. The paper hypothesizes a ubiquitination mode of action where the substrate does not disengage, yet this remains unproven.

8. The reported three-digit resolutions (as in Extended Data Table 1) and distances (Figure 4b) lack practicality. Reducing these to a maximum of two and recognizing that the distances cannot be measured with such precision would be appropriate.

9. Lines 202–210, related to Figure 3e, f, veer into speculative territory and would be better placed in the discussion section.

10. Lines 226–230 also delve into speculation without evidence that the substrate does not disengage following each ubiquitination before re-engaging. This should be reserved for discussion.

11. Trans ubiquitination is suggested (Lines 240–255), but this is not verified experimentally. While challenging to demonstrate, this might be more suitably addressed in the discussion.

12. Mutations mentioned in the abstract are, in some instances, only addressed in the discussion and are not demonstrated.

Minor Comments:

13. The helices naming is unclear; "Nhelix" and "loop-helix" appear to be synonymous but are both used. Consider selecting one term. Likewise, consider more transparent naming, e.g., renaming "linker helix" to "E6 linker helix."

14. Please revise the structure of figure citations in the text, e.g., Figures 1d, e are mentioned after 2a, b. This inconsistency also appears with Figures 2e, f, and others.

15. Color coding is inconsistent across figures.

16. Data (e.g., extfigure9) should not be introduced in the discussion section.
17. The numbers in Extended Data Figure 1 do not align. The sum of the particle numbers from the final reconstructions is only half of the total prior to refinement. It's unclear at what refinement stage such a significant number of particles were excluded.
18. A close-up of E6 in Figure 1c would be more insightful if presented in the context of the overall structure.
19. The manuscript contains numerous typos and missing words, warranting a thorough check (e.g., "kv" should be "kV").
20. The utility of structure close-ups is diminished without dashed lines to indicate distances.
21. The Att state presented in the individual figures is unclear; please specify.
22. In Figure 2, the gel and blot are not scaled similarly, reducing comparability. Please adjust or add a marker for reference.
23. Overall the Figures could be better organized and be made more accessible.

We would like to express our gratitude to the editor and the two reviewers for their time in evaluating our paper. As you will see from our point-by-point response below, we have been thorough in our attempts to address all the points raised by the reviewers. We have revised the paper accordingly.

To facilitate your navigation of this response statement, the referees' comments are in **black**, and our responses are in **blue**.

Referees' comments:

Reviewer #1 (Remarks to the Author):

The work presented by Wang et al. consists of results from structural analyses performed by cryo-EM on the E6AP E3 ubiquitin ligase, both in the free state and bound to the HPV E6 oncoprotein. Whereas free E6AP is essentially observed in a monomeric state, the E6/E6AP complex dimerizes. The resulting E6/E6AP heterotetramer is held in place by inter-protomer contacts involving mainly residues of E6AP within the alpha1-helix but also of the N-helix of E6AP. In the tetramer, the E6 protein from one protomer is in close proximity with the C-lobe of the E6AP HECT domain from the other protomer. Different conformational states of the E6/E6AP tetramer are observed, which are clustered into two main conformational families: the "Attached" and "Detached" conformations. The "Attached" conformations display closer contacts between E6 and the C-lobe as compared to the "Detached" conformations. Such tetrameric organization suggests an inter-protomer mechanism for E6 ubiquitination by E6AP. The authors then used the different E6/E6AP structures to dock the core domain of p53. Only one structure from the Detached family can accommodate p53 without clashes. Base on this model, p53 ubiquitination would also happen through a trans-protomer mechanism but would require a "Detached" conformational state.

In summary, the authors provide very elegant structural data in support of long-standing issue in the field. My main concerns are on the results from biochemical/functional analyses in support of the trans-protomer mechanism.

Re: We are very grateful for the reviewer's positive comments regarding our study.

Major issues:

Fig 2. Panel f: I do not see clear differences in E6 band intensities between wt E6 and E6 D32A/Y50A/R148A. From this experiment it seems to me that suppression of the E6/C-lobe contacts has no effect on E6 ubiquitination. Panel g: the details of this experiment are not described. Which is the protein with the HA tag? I guess it is ubiquitin...How are the authors sure that what they see is E6 ubiquitination and not E2 ubiquitination for example?

Re: We thank the reviewer for the comment and apologize for any misunderstanding due to the lack of a M.W. marker and protein identification in Fig. 3f (related to previous Fig. 2g). The two gels in Fig. 3e (related to previous Fig. 2f), right, and Fig. 3f were generated using the same loading samples and stained with Coomassie and immunoblotting, respectively. The ubiquitin moiety contains the HA tag. We agree that the intensity of the E6 band in the E6 mutant lane stained with Coomassie in Fig. 3e (right) is not significantly different from that of the E6 wt. However, the immunoblotting assay shows a clear decrease in polyubiquitination in the mutant sample (Fig. 3f), and this is more specific than Coomassie staining. Further, we confirmed this

to be MBP-E6-Ub_(n). Finally, regarding the reviewer's final comment, in another ubiquitination assay (Supplementary Fig. 6), UbcH7(E2) was unable to be ubiquitinated by E6AP even with in the presence of the MBP-tagged E6 (MBP-E6). Also, the location of lowest band in the WB gel is much larger than E2-Ub and is roughly at the level for MBP-E6-Ub, which may provide additional evidence to support E6 polyubiquitination in the assay. We have modified the figure and the corresponding text in the revised manuscript (page 8, line 213-219).

Fig5, panel h: here the E6 mutants tested affect residues located at the E6/E6AP interface adjacent to the LxxLL motif region. Results show that these E6 mutants have reduced p53 ubiquitination activities. However, they do not prove that the trans-protomer mechanism is required for p53 ubiquitination. To show this, they would have to make mutants that impair E6AP dimerization through the alpha-1 helix of E6AP and test them for p53 ubiquitination. This is an important experiment. Indeed, one should also keep in mind that full-length p53 is a tetramer and that this could have an impact on the p53 ubiquitination mechanism.

Re: We thank the reviewer for this valuable suggestion. To further support our data, we constructed an E6AP mutant R505P, which is a known loss-of-function mutation associated with Angelman syndrome. The mutation is located on the α 1-helix, the region known to mediate the dimerization of two protomers (Figs. 4a). The dynamic light scattering (DLS) assay results show that the polydispersity (pd) of the mutant is significantly higher than that of the wild-type, indicating the E6AP mutant/E6 complex is much less stable than the E6AP wildtype/E6 complex (Fig. 4b). Consistently, the ubiquitination assay results show that the mutation significantly reduces the ubiquitination of p53 (Fig. 4c, d). These results, coupled with our structural analyses, suggest that p53 is ubiquitinated by the E6AP/E6 complex *in trans*. We have added the corresponding text to the revised manuscript (page 9, line 244-252).

Minor issues:

-Please clarify in the text what are the E6 constructs used in the different experiments. Is it wt HPV16 E6?

Re: We used the E6 4C/S mutant in all our experiments; this was because previous studies (Zanier K., Science, 2013; Martinez-Zapien D., Nature, 2016) have shown that this mutation leads to significantly reduced aggregation mediated by disulfide formation, while maintaining a potency comparable to that of the E6 wild type in E6AP-mediated p53 ubiquitination. We have revised the method section to clarify this (page 15, line 402-406).

-Please refer to Sailer C et al Nat Commun 2018. In this paper E6 has already been shown to enhance E6AP activity by increasing oligomerization.

Re: We have incorporated the viewpoint presented by Sailer et al. in our manuscript (page 13, line 352-354).

-There are several paragraphs that require rephrasing/corrections. Please revise the text.

Re: Following the reviewer's suggestion, all authors have revised the text thoroughly.

Reviewer #2 (Remarks to the Author):

Wang et al. present the structure of the HECT-E3 E6AP in complex with the viral protein E6, revealing a complex that incorporates two copies of each protein. This observation raises pivotal mechanistic questions regarding the implications of such dimerization, which the authors duly explore. Given the recent surge in elucidating HECT ligase structures and insights derived through cryo-EM, the manuscript is particularly timely. Nonetheless, I posit that some of the conclusions drawn are not wholly substantiated by the data presented, and that the structural work may benefit from further refinement. Consequently, I suggest that the authors consider revising their manuscript, taking into account the following concerns and recommendations.

Re: We are very grateful to the reviewer for their positive comments on our study, and appreciate the reviewer's valuable suggestions for improving our manuscript.

1. I appreciate the authors sharing their maps and models, many of which are of commendable quality. Nevertheless, some issues of overfitting and anisotropy are evident in the map, as demonstrated through the excessively high-resolution estimates. Att1 certainly does not achieve sub-3Å resolution. A possible explanation might be that the dataset contains a considerable amount of noise. The authors initially selected 16 million particles, utilizing only a small fraction following 2D classification, indicating a significant number of false positives during picking. I recommend exploring alternative picking tools, such as cryoLO or Topaz, to enhance this step, as 2D classification is not effective at removing noisy particles. A reprocessing of the data, potentially employing DeepEMhancer, may further enhance map quality.

Re: We appreciate the reviewer's commendation of the map and model quality. The resolution of Att1—like that of other states—was determined using the gold-standard Fourier shell correlation (overall) and Resmap for the local distribution analysis. We agree with the reviewer the map might have an anisotropy issue. By following the reviewer's advice, we utilized DeepEMhancer to process the maps. This generated reconstructions with improved overall map quality, among which the Att1 is comparable with a sub-3 Å resolution reconstruction as compared with other recently published ones (Wang F., *Structure*, 2023; Hehl L., *Nat Chem Biol*, 2023). The revised manuscript now includes these updated maps (Fig. 1a, b and Supplementary Figs. 1, 3). We thank the reviewer for the advice regarding employing DeepEMhancer, as this has improved the quality of our manuscript.

During data processing, we picked more particles with lower thresholds in order to capture a higher number of possible states of the E6AP/E6 complex. We apologize for the unclear description of the data processing workflow in Supplementary Fig.1d. We actually removed the bad particles by both 2D and 3D classification. For clarity, we now include more information in the revised manuscript. Further, per the reviewer's suggestion, we re-picked particles using Topaz and found over 80% particles picked using Relion are overlapped with that picked using Topaz (Response Figure 1). As such, we have retained our original procedure for particle picking.

Response Figure 1. Overlap of Relion-selected and Topaz-selected particles. 500 micrographs were selected randomly to pick particles. The ratios of the overlapping particles (Overlapping_num) selected by Relion and Topaz in each micrograph are presented relative to particles (All particle_num) selected by Relion alone.

2. An issue arises when observing that the maps and models are misaligned when visualized in Chimera or Coot, which could potentially produce inaccurate statistics. The root cause of this misalignment warrants investigation.

Re: We apologize for the mistake. Before sending the maps and models to the reviewer, we have checked the alignments between each map and its corresponding model. It is likely that we accidentally saved the somehow shifted model of Att1 during the process. We have re-uploaded the updated model of Att1 into the Protein Data Bank.

3. The model of the monomer map should be restricted to docking and not subjected to building and refinement due to the inadequate quality of the map. Refinement may introduce more errors than it ameliorates.

Re: We have removed the model of the monomer map and docked the E6AP model of the E6AP/E6 complex into the monomer map to analyze the structural changes of E6AP caused by E6 binding (Fig.6a, b). We have removed model-building content from the manuscript and modified the refinement statistics in Supplementary Table 1.

4. While the paper employs C2 Symmetry, it also explores movements, which appear to lack symmetry, as evidenced by the smeared-out density of the tower. A consideration of C1 processing might provide additional insights.

Re: The smeared-out density of the tower could arise either from symmetry averaging or from its intrinsic dynamicity. Following the reviewer's suggestion, we have re-processed the data of the Att1 state using C1 symmetry. We found that the quality of the C1 Att1 map is similar to that of the 2.6-Å C2 map, but with a compromised resolution of 2.8-Å. It is worth noting that the tower of the C1 map is not better than that of the C2 map (Respond Figure 2). The smeared-out density is more likely due to intrinsic dynamics rather than the C2 averaging.

Response Figure 2. Local resolution (a) and the overall resolution (b) of C2 and C1 symmetry-processed maps.

5. Although the paper posits a sequential order of movements with "towers" modulating the pivot point of the alpha helix, the data does not conclusively support this. Moreover, the utilization of discrete structures on apparent continuous movements is peculiar. Implementing a modern flexibility analysis method, such as 3DVA, 3DFlex, or cryoDRGN, might yield deeper insights into the dynamics.

Re: We thank the reviewer for the comment. We did not intend to imply that the dynamicity of the pivot point of the alpha helix is modulated by the "towers"; we used this concept primarily to explain the structural changes between the different states of the complex. To avoid any confusion, we have modified our statements in the revised manuscript (page 7, line 177-181). Furthermore, following the reviewer's suggestion, we used the particles in the three attached states (approximately 419,000 particles; Supplementary Fig.1d) and applied 3DVA to analyze the structural dynamics (Supplementary video 1; page 7, line 181-186). The result displayed a series of conformations of the complex and illustrated continuous movement between the two protomers. The three structures in the Att state presented in this study represent different stages of the structural continuum of E6AP/E6 binding, as they exhibit similar conformational changes compared to those revealed by 3DVA. This data has been included in the revised manuscript, and greatly improves the quality of our results.

6. While the authors posit that dimerization controls activity, this is not empirically demonstrated. The reversibility of dimerization is not addressed, nor are the specific E6AP regions required for dimerization experimentally explored. Implementing site-directed mutagenesis and utilizing SLS or EM might provide necessary confirmation of specific interface involvement.

Re: We thank the reviewer for the comment. We have constructed an E6AP mutant R505P, a known loss-of-function mutation associated with Angelman syndrome (page 9, line 244-251). The mutation is located on the extended region (residues 503-514) of the α 1-helix, which mediates the dimerization of the two protomers (Figs. 4a and 6b). The dynamic light scattering (DLS) assay results show that the polydispersity (pd) of the mutant is significantly higher than that of the wild-type, indicating that the E6AP mutant /E6 complex is much less stable than the E6AP/E6 complex (Fig. 4b). Consistently, the ubiquitination assay results showed that the mutation significantly reduces the ubiquitination of p53 (Fig. 4c, d). These results indicate that the extended α 1-helix is indeed required for the dimerization and the p53 ubiquitination.

During processing of the E6AP dataset, we observed that two of the 2D averages exhibited features reminiscent of the dimerized E6AP configuration (Supplementary Fig. 11), even though E6AP mainly exists as a monomer. This is consistent with previous studies (Nuber U., Eur J Biochem, 1998; Sailer C., Nature communications, 2018; Ronchi V., J Biol Chem, 2014) that suggest that E6AP may exist as a reversible dimer. To avoid any overstatement of the results, we have removed the statement about reversibility.

7. The paper hypothesizes a ubiquitination mode of action where the substrate does not disengage, yet this remains unproven.

Re: We apologize for any confusion. The earlier Fig. 6e may evoke the idea of ubiquitin chain elongation on the substrate without disengagement. We have revised the manuscript (page 13, line 346-350) and Fig.6e to eliminate any possible confusion.

8. The reported three-digit resolutions (as in Extended Data Table 1) and distances (Figure 4b) lack practicality. Reducing these to a maximum of two and recognizing that the distances cannot be measured with such precision would be appropriate.

Re: We thank the reviewer for the suggestion. The distance and resolution accuracies have been reduced to two digits (Supplementary Fig. 5, related to previous Fig. 4b; Supplementary Table 1).

9. Lines 202–210, related to Figure 3e, f, veer into speculative territory and would be better placed in the discussion section.

Re: We agree with the reviewer on this point. We have moved the statement to the Discussion section in the revised manuscript (page 12, line 339-346; Supplementary Fig. 13, related to previous Fig. 3e, f).

10. Lines 226–230 also delve into speculation without evidence that the substrate does not disengage following each ubiquitination before re-engaging. This should be reserved for discussion.

Re: We agree with the reviewer on this point. We have moved the statement to the Discussion section in the revised manuscript (page 13, line 346-350).

11. Trans ubiquitination is suggested (Lines 240-255), but this is not verified experimentally. While challenging to demonstrate, this might be more suitably addressed in the discussion.

Re: Our structural analysis has suggested p53 cannot be ubiquitinated by E6AP within one same protomer. As we have answered to question #6, we carried out new experiments (Fig. 4b-d; page 9, line 244-252). The results from the new experiments indicate that the extended α 1-helix is required for the dimerization, and that the p53 is ubiquitinated by E6AP *in trans* in the context of dimer of E6AP/E6 protomer.

12. Mutations mentioned in the abstract are, in some instances, only addressed in the discussion and are not demonstrated.

Re: We understand the reviewer's concern here. As we answered to question #6, we constructed a mutant R505P of E6AP, which is a known loss-of-function mutation associated

with Angelman syndrome, and the mutation destabilizes the E6AP/E6 complex and decreases the E6AP activity for p53 ubiquitination. We have also revised the abstract (page 2, line 34).

Minor Comments:

13. The helices naming is unclear; "Nhelix" and "loop-helix" appear to be synonymous but are both used. Consider selecting one term. Likewise, consider more transparent naming, e.g., renaming "linker helix" to "E6 linker helix."

Re: Thank you for pointing this out. "N-helix" and "loop-helix" are now both referred to as "N-helix" (page 5, line 128; page 6, line 163 and 165). We have renamed "linker helix" to "E6 linker helix" within our manuscript (page 5, line 114; page 6, line 164; page 10, line 259 and 270).

14. Please revise the structure of figure citations in the text, e.g., Figures 1d, e are mentioned after 2a, b. This inconsistency also appears with Figures 2e, f, and others.

Re: We have revised the structure of figure citations in the text.

15. Color coding is inconsistent across figures.

Re: We have changed the color coding to be consistent across figures. This is also clearly stated in the figure legends.

16. Data (e.g., extfigure9) should not be introduced in the discussion section.

Re: We have moved the data to the Results section (page 11, line 306-309; Supplementary Fig. 11, related to previous Supplementary Fig. 9).

17. The numbers in Extended Data Figure 1 do not align. The sum of the particle numbers from the final reconstructions is only half of the total prior to refinement. It's unclear at what refinement stage such a significant number of particles were excluded.

Re: We thank the reviewer for this comment and apologize for the mistake made in the Supplementary Fig. 1d. There is a round of 3D classification procedure missing after the detached map is initially separated, which caused the misaligned particle numbers. Now we have corrected the information in the revised figure.

18. A close-up of E6 in Figure 1c would be more insightful if presented in the context of the overall structure.

Re: We thank the reviewer for this suggestion. We have modified this figure (Fig. 1e, related to previous Fig. 1c).

19. The manuscript contains numerous typos and missing words, warranting a thorough check (e.g., "kv" should be "kV").

Re: We have checked the entire manuscript and corrected typos and missing words (page 4, line 97; page 16, line 443).

20. The utility of structure close-ups is diminished without dashed lines to indicate distances.

Re: We have added dashed lines to indicate distances (Fig. 2).

21. The Att state presented in the individual figures is unclear; please specify.

Re: We have revised the figures to specify the Att state of the structure (Figs. 1, 5 and 6).

22. In Figure 2, the gel and blot are not scaled similarly, reducing comparability. Please adjust or add a marker for reference.

Re: We thank the reviewer for this comment, and we have added a protein marker to Fig. 3f (related to previous Fig. 2g).

23. Overall the Figures could be better organized and be made more accessible.

Re: Following the reviewer's suggestion, we have modified and re-organized our figures.

Reviewers' Comments:

Reviewer #1:

Remarks to the Author:

In the revised version of the manuscript, the authors have answered to my concerns.

Major issue 1 – related to biochemical experiments of former Fig. 2 / new Fig 3e and f.

In the new version of the manuscript, the authors show molecular weight markers and have performed an additional experiment to rule out E2 ubiquitination of E6AP. This is good. However, the conclusion drawn from these data are too strong: "These results suggest that interactions between the E6AP C-lobe and E6 are essential for the donation of ubiquitin from E6AP to its substrate (page 8)". The authors should rather say that "interactions between the E6AP C-lobe and E6 contribute to the donation of ubiquitin...".

Major issue 2 – related to experimental demonstration of the trans-protomer mechanism.

In the new version of the manuscript, the authors have analyzed a pathogenic mutant of E6AP which bears an R505P substitution in the alpha-1 helix. The results obtained support the trans ubiquitination mechanism.

Reviewer #2:

Remarks to the Author:

The authors have made significant improvements to the manuscript, incorporating many of the suggestions previously provided. This manuscript represents a valuable contribution to the field and merits consideration for publication in Nature Communications. However, certain critical aspects, particularly regarding the functional claims about trans ubiquitination, require further clarification to prevent misinterpretation and overinterpretation. I would like to highlight a few points that need attention:

1. The ubiquitination assay and its quantification in Figure 4 have issues: The current method for assessing ubiquitination, does not provide a clear measure due to overlapping bands and ambiguous quantification. I recommend utilizing an Anti-flag Western blot to improve clarity and specificity. Additionally, the mutation's impact on polydispersity and stability is not convincingly demonstrated, which could explain the observed decrease in ubiquitination activity. A more definitive method, such as mass photometry, should be employed for a clearer characterization. Also use of fluorescent ubiquitin may help to clarify the ubiquitination kinetics.

2. The manuscript does not convincingly demonstrate ubiquitination in trans, especially in the context of the mutation study. The observed ubiquitination in the absence of dimer formation suggests a cis mechanism rather than trans. This claim requires either stronger evidence or a significant revision in the narrative to reflect the data accurately.

3. The discussion section suffers from a lack of fluidity due to the direct insertion of a suggested paragraph. To improve readability and coherence, I recommend a thorough revision of this section.

Minor comment:

4. Although there are still minor technical issues with the manuscript's structure, they are not sufficient to preclude acceptance. Nevertheless, the authors approach in picking and particle sorting does not remove empty noise particles sufficiently. I invite the authors to perform the following an exercise: I recommend conducting an additional experiment to illustrate the limitations of classification in removing empty particles. Specifically, by introducing 10% synthetic noise particles into the dataset, the authors will likely observe that a significant portion of noise remains, regardless of classification accuracy. This underscores the importance of cautious interpretation and the need for setting appropriate thresholds in all picking tools to avoid overfitting, despite the application of gold standard FSC criteria. This suggestion is meant to enhance the authors understanding of cryo EM rather than as a direct critique of the manuscript.

In summary, while the manuscript has undergone substantial improvement and presents valuable structural insights, addressing these specific points will significantly enhance its scientific rigor and clarity. I look forward to seeing these revisions, which I believe will make a substantial contribution to the field upon publication in Nature Communications.

We again express our gratitude to the editor and the two reviewers for their time in evaluating our paper. As you will see from our point-by-point response below, we have been thorough in our attempts to address all the points raised by the reviewers. We have revised the paper accordingly.

To facilitate your navigation of this response statement, the referees' comments are in **black**, and our responses are in **blue**.

Referees' comments:

Reviewer #1 (Remarks to the Author):

In the revised version of the manuscript, the authors have answered to my concerns.

Major issue 1 – related to biochemical experiments of former Fig. 2 / new Fig 3e and f. In the new version of the manuscript, the authors show molecular weight markers and have performed an additional experiment to rule out E2 ubiquitination of E6AP. This is good. However, the conclusion drawn from these data are too strong: “These results suggest that interactions between the E6AP C-lobe and E6 are essential for the donation of ubiquitin from E6AP to its substrate (page 8)”. The authors should rather say that “interactions between the E6AP C-lobe and E6 contribute to the donation of ubiquitin...”.

Re: The conclusion has been changed as the reviewer suggested (page 8, line 217-218).

Major issue 2 – related to experimental demonstration of the trans-protomer mechanism.

In the new version of the manuscript, the authors have analyzed a pathogenic mutant of E6AP which bears an R505P substitution in the alpha-1 helix. The results obtained support the trans ubiquitination mechanism.

Re: Thanks for the reviewer's recognition for our result.

Reviewer #2 (Remarks to the Author):

The authors have made significant improvements to the manuscript, incorporating many of the suggestions previously provided. This manuscript represents a valuable contribution to the field and merits consideration for publication in Nature Communications. However, certain critical aspects, particularly regarding the functional claims about trans ubiquitination, require further clarification to prevent misinterpretation and overinterpretation. I would like to highlight a few points that need attention:

1. The ubiquitination assay and its quantification in Figure 4 have issues: The current method for assessing ubiquitination, does not provide a clear measure due to overlapping bands and ambiguous quantification. I recommend utilizing an Anti-flag Western blot to improve clarity and specificity. Additionally, the mutation's impact on polydispersity and stability is not convincingly demonstrated, which could explain the observed decrease in ubiquitination activity. A more definitive method, such as mass photometry, should be employed for a clearer characterization. Also use of fluorescent ubiquitin may help to clarify the ubiquitination kinetics.

Re: Following the reviewer's suggestion, we have utilized the Anti-flag Western blot to analyze ubiquitination (Supplementary Fig. 8a, b in the revised manuscript). The results from the two

methods are essentially similar (Fig. 4c-d and Supplementary Fig. 8a, b). Polydispersity (pd) has been used to indicate the thermal stability of protein complex (Shiba, K., *Anal Sci*, 2010; Kwan, T., *Front Mol Biosci*, 2022). Our DLS assay shows clearly that the pd of the mutant E6AP/E6 complex is significantly higher than that of the wild type complex, indicating that the mutant complex has a decreased stability than that of the wildtype complex. Unfortunately, due to instrument unavailability, we are unable to perform the mass spectrometry assay.

2. The manuscript does not convincingly demonstrate ubiquitination in trans, especially in the context of the mutation study. The observed ubiquitination in the absence of dimer formation suggests a cis mechanism rather than trans. This claim requires either stronger evidence or a significant revision in the narrative to reflect the data accurately.

Re: The mutation R505P in E6AP decreases rather than abolishes the dimer formation. Therefore, the observed ubiquitination should come from the dimer of the mutant E6AP/E6 protomer rather than the mutant E6AP/E6 heterodimer. Following the reviewer's suggestion, we have removed the claim about trans ubiquitination and revised the narrative for the data (page 4, line 86-87; page 8, line 202-203; page 9, line 251-252).

3. The discussion section suffers from a lack of fluidity due to the direct insertion of a suggested paragraph. To improve readability and coherence, I recommend a thorough revision of this section.

Re: The discussion section has been modified to improve the fluidity (page 12, line 336-339; page 13, line 353-356).

Minor comment:

4. Although there are still minor technical issues with the manuscript's structure, they are not sufficient to preclude acceptance. Nevertheless, the authors approach in picking and particle sorting does not remove empty noise particles sufficiently. I invite the authors to perform the following an exercise: I recommend conducting an additional experiment to illustrate the limitations of classification in removing empty particles. Specifically, by introducing 10% synthetic noise particles into the dataset, the authors will likely observe that a significant portion of noise remains, regardless of classification accuracy. This underscores the importance of cautious interpretation and the need for setting appropriate thresholds in all picking tools to avoid overfitting, despite the application of gold standard FSC criteria. This suggestion is meant to enhance the authors understanding of cryo EM rather than as a direct critique of the manuscript.

Re: As we indicated previously, we conducted many rounds of 2D and 3D classifications to eliminate bad and empty particles. We resolved 5 conformation states of the E6AP/E6 in this study, and the map quality of four conformation states was commended by the reviewer in the first-round review. The Att1 conformation state was also improved by using the DeepEMhancer as the reviewer suggested. The structural features of the five conformation states corroborate to each other, which effectively alleviates the inappropriate interpretation. We fully believe the limitation of the current classification methods. However, due to time and computer resource constraints, we are not able to perform the test the reviewer suggested.

In summary, while the manuscript has undergone substantial improvement and presents valuable structural insights, addressing these specific points will significantly enhance its scientific rigor and clarity. I look forward to seeing these revisions, which I believe will make a substantial contribution to the field upon publication in Nature Communications.

Re: We appreciate the reviewer's valuable advice, which we believe has significantly improved the quality of our manuscript. We have done our best to revise the manuscript. We hope our responses meet the reviewer's satisfaction.